# JOINT TRAINING DOES NOT TRANSFER INFORMATION BETWEEN EEG AND IMAGE CLASSIFIERS

## ABSTRACT

Caution is necessary with machine-learning methods, and especially computer-vision methods, to support brain processing claims from neuroimaging data. A recent paper (Palazzo et al., 2021) proposes (i) a joint-training process that does not use class information and (ii) a bidirectional transfer of (a) image information to an EEG classifier and (b) brain-activity information to an image classifier, such that the joint embedding includes the shared image and brain-activity information. These claims cannot be maintained: the training process is initialized with class information, and joint training with EEG degrades rather than improves the performance of the image encoder. Moreover, theoretical solutions exist that entail no transfer beyond class information in the joint embedding space.

## 1 INTRODUCTION

A recent paper (Palazzo et al., 2021) proposes a method for jointly training two neural networks, one to map images to encodings and the other to map EEG data from subjects viewing those images to encodings, so that the encodings for the images are similar to those from the EEG data. Specifically, they claim that

I) *class labels are not used anywhere in the equation. This makes sure that the resulting embedding does not just associate class discriminative vectors to EEG and images, but tries to extract more comprehensive patterns that explain the relations between the two data modalities.* (Palazzo et al., 2021, § 3 ¶ 5) and

II) *our multimodal approach learns a joint brain-visual embedding and finds similarities between brain representations and visual features* (Palazzo et al., 2021, § 1 ¶ 2).

They use the resulting encodings for EEG classification, image classification, saliency detection, and producing activation maps that decode brain representations. Central are the additional implicit claims

III) that the joint embedding space encodes both information about the images and subjects' brain activity from viewing the images and

IV) that the joint-training process bidirectionally transfers information from the image encoders to the EEG encoders, and information from the EEG encoders to the image encoders.

Other work (Li et al., 2021; Ahmed et al., 2022) questioned these claims due to confounds in the data (Spampinato et al., 2017), which was collected with a block design with all stimuli of a given class presented to subjects in close proximity and the training and test sets containing samples from the same block. Since EEG data contains temporal drift, this drift is classified rather than stimulus-related brain activity. Li et al. (2021) and Ahmed et al. (2022) note that it is critical to remove the confound by breaking the correlation between temporal drift and stimulus class by randomizing stimulus presentation order. Li et al. (2021) and Ahmed et al. (2022) demonstrated that numerous classifiers, including the one used in Palazzo et al. (2021), lose high classification accuracy seen on the data from Spampinato et al. (2017) used in Palazzo et al. (2021), dropping to chance with nonconfounded data properly collected from randomized trials.

Here, we focus on a separate issue. We demonstrate that their method itself does not exhibit claims I–IV, whether applied to confounded or nonconfounded data. We contribute the novel observations that independently refute the claims in Palazzo et al. (2021) beyond Li et al. (2021) and Ahmed et al. (2022):

A) Prior to joint training, the pretrained image encoders contain close to perfect class information, and likely very little information other than class information. This calls claim I above into question.

B) The models used as EEG and image encoders, and the loss function used to jointly train those encoders, have the representational capacity to memorize one-hot EEG and image encodings that minimize the loss function on the training set. These models can be found with separate training. Thus it is unlikely that joint training transfers information from the image encoders to the EEG encoders or vice versa, or includes anything other than class information. This calls claims II–IV above into question.

We demonstrate that their method doesn't support their claims, even when applied to data that remedies the confound. Since their claims are contingent upon production of joint embeddings that include both image and brain-activity information, this calls all claims in Palazzo et al. (2021) into question.

Beyond this, we further demonstrate

C) that joint training appears to (1) use the class information included in the pretrained image encoders to train the EEG encoder as a classifier, but (2) otherwise degrades performance of the image encoder as a classifier and

D) that while the EEG encoders so trained can perform above chance on confounded data, they critically perform at chance on nonconfounded data.

Thus their joint-training method is ineffective.

Here, we limit our analyses and discussion to claims A–D, in so far as they refute claims I–IV. This does not hinge on the confound reported in Li et al. (2021) and Ahmed et al. (2022); in fact, we perform the same analyses both on confounded and nonconfounded data, obtaining the same results. Our logical arguments all are based on the information content in the encodings at various stages of various kinds of training, where information content is measured by principal component analysis (PCA). It is expressly not based on absolute classification accuracy, *i.e.*, how well the classifiers perform. We are concerned with relative classification accuracy of various reconstructions of the encodings with limited sets of components derived through PCA as a way of assessing information content of the encodings.

Palazzo et al. (2021) use the EEG and image encodings produced by jointly trained EEG and image encoders to produce saliency and activation maps. The validity of these maps hinges on the claim that these encodings contain both visual and brain-activity information. We demonstrate this to be false, calling the validity of these maps into question. Li et al. (2021, § 5.4) already questioned this validity both on methodological grounds and as a result of the confound. Our results here offer additional independent reason to question the validity of these maps.

We make no claim that it is difficult or impossible to perform the central task attempted by Palazzo et al. (2021), namely to perform joint training of EEG and image encoders to transfer image information to the EEG encoder and brain-activity information to the image encoder and yield a joint embedding that contains both image and brain-activity information, without providing class information. It might be possible to perform this task with a different method, or even with the same method but with different data. Our sole claim is that their experiments and analyses do not demonstrate that they have performed this central task as stated by claims I–IV made in Palazzo et al. (2021).

## 2 METHOD

We jointly train the EEG encoder (EEGChannelNet, Palazzo et al., 2021) with each of the same image encoders employed by Palazzo et al. (2021): Inception v3 (Szegedy et al., 2016), ResNet-101 (He et al., 2016), DenseNet-161 (Huang et al., 2017), and AlexNet (Krizhevsky et al., 2012). Our analyses report results for all four image encoders, each analyzed two ways: without pretraining ('no pretraining') and pretrained on the ILSVRC 2012 training set ('pretraining'). As we evaluate just the encodings, we do not postpend classifiers to these encoders. Thus these encoders produce 1000-element output vectors, not 40-element ones.

We train the encoders in two ways: (1) joint training as described in Palazzo et al. (2021), with triplet loss, and (2) separate training individually with MSE loss against one-hot class encodings.

After separate training, we run a single epoch of joint training, without updating the weights, just to compute the triplet loss. We report three kinds of results: 'before' is before any joint or separate training, 'after joint' is after joint training, and 'after separate' is after separate training.

We repeat this analysis for three different EEG datasets: 'their' refers to the data from Spampinato et al. (2017) used by Palazzo et al. (2021), 'block' refers to subject 6 first image block run from Li et al. (2021), and 'randomized' refers to subject 6 image rapid-event run from Li et al. (2021). 'Their' and 'block' are confounded; 'randomized' is not.

We use the encoders, trained just on the training sets, to produce encodings on the validation and test sets for all splits and pool the results. For 'block' and 'randomized' data, the validation and test sets are a disjoint cover of the dataset so each sample has exactly one encoding. Critically, the data from Spampinato et al. (2017) does not have this property. We perform two analyses on these encodings. First, we perform principal-component analysis (PCA) and reconstruct the encodings two ways: one with just the 'top 40' components and one with just the 'bottom 960' components and report the fraction of variance in the encodings explained by the top 40 components. Second, for both the raw encodings ('original') and the reconstructed encodings ('top 40' and 'bottom 960'), we compute the accuracy when using the encodings for classification, without postpending any classifier (a $1000 \rightarrow 40$ FC layer followed by a softmax as done by Palazzo et al. 2021). Since the 40 classes employed by Palazzo et al. (2021) are a subset of the 1000 classes in ILSVRC 2012, and the image stimuli used by Spampinato et al. (2017) to elicit EEG response are a subset of the training images in ILSVRC 2012, one can read off the class label directly from the 1000-element encoding produced by either the EEG or image encoders by choosing the index of the maximal element. We do this two ways. The first ('1000 classes') considers maximal elements outside of the 40 classes to be misclassification. The second ('40 classes') ignores elements outside of the 40 classes and only computes the maximal element among the 40 classes.

We also analyze the value of the loss function: for the triplet loss after both 'joint' and 'separate' training, on all four image encoders, both with and without 'pretraining,' and all three datasets, and also the MSE loss when separately training the 'EEG' and 'image' encoders. These losses are computed per sample and per element of the 1000-element encoding, averaged over samples and splits.

The appendix contains more details of our method.

## 3 RESULTS

Table 1 shows the fraction of variance explained by the top 40 principal components of the encodings produced for both modalities ('EEG' and 'image') for all image encoders and datasets, 'before' joint or separate training, 'after joint' training, both with and without 'pretraining,' and 'after separate' training.[1]

- The top 40 principal components explain a large portion of the image-encoding variance ($\geq 63.9\%$) before training (column v; Table 6[2]). This together with Tables 2 and 3 implies that the pretrained image encoders produce encodings that contain (primarily) class information on all data. This supports claim A and calls claim I into question.
- After joint training with pretraining on their data, the top 40 principal components explain almost all ($\geq 94.9\%$) of the variance in the image encodings (column vi, for rows i–iv; Table 7). This implies that the image encoders jointly trained on their data produce encodings with little more than class information.
- After separate training, the top 40 principal components explain almost all of the variance ($\geq 96.8\%$) in the image encodings (column viii; Table 8). This implies that the separately trained image encoders produce encodings with little more than class information on all data.

---

[1]Note that while the entries in Table 1 are percentages, they are not classification accuracies. Explained variance measures how close a reconstruction of the encodings using just the top 40 components is to the original encodings. High values indicate that the reconstruction is close to the original encodings because the bulk of the information in the encodings resides in the top 40 components.

[2]Here and throughout, tables numbered six and higher refer to those in the appendix. These contain variants of the five main tables with the indicated columns and rows highlighted in color.

Table 1: Explained variance (%) in the top 40 principal components of the encodings, 'before' joint or separate training, 'after joint' training, both with and without 'pretraining,' and 'after separate' training. Rows i–iv 'their,' rows v–viii 'block,' and rows ix-xii 'randomized'. Rows i, v, and ix Inception v3. Rows ii, vi, and x ResNet-101. Rows iii, vii, and xi DenseNet-161. Rows iv, viii, and xii AlexNet.

| | EEG | | | | image | | | |
| | before | after joint | | after separate | before | after joint | | after separate |
| | | pretraining | no pretraining | | | pretraining | no pretraining | |
| | i | ii | iii | iv | v | vi | vii | viii |
|---|---|---|---|---|---|---|---|---|
| i | 44.7 | 37.2 | 15.3 | 99.5 | 64.1 | 94.9 | 98.9 | 100.0 |
| ii | 46.5 | 36.2 | 17.7 | 99.5 | 82.0 | 98.9 | 98.2 | 96.8 |
| iii | 48.2 | 25.6 | 22.7 | 100.0 | 75.4 | 97.4 | 90.2 | 100.0 |
| iv | 46.3 | 21.7 | 18.5 | 99.5 | 82.3 | nan | 100.0 | 99.7 |
| v | 83.3 | 75.9 | 69.4 | 42.3 | 63.9 | 57.4 | 98.3 | 99.9 |
| vi | 85.4 | 79.8 | 72.8 | 41.9 | 81.9 | 99.5 | 99.0 | 99.7 |
| vii | 84.0 | 79.3 | 76.3 | 42.9 | 75.3 | 69.4 | 94.3 | 99.5 |
| viii | 84.4 | 78.9 | 73.2 | 40.0 | 82.2 | 78.9 | 99.9 | 98.9 |
| ix | 62.1 | 40.5 | 38.3 | 19.3 | 63.9 | 60.2 | 98.1 | 99.9 |
| x | 61.9 | 52.2 | 43.6 | 19.7 | 81.9 | 99.1 | 98.8 | 99.8 |
| xi | 63.5 | 48.5 | 46.5 | 19.6 | 75.3 | 70.8 | 93.9 | 99.7 |
| xii | 63.6 | 49.5 | 41.3 | 19.2 | 82.2 | 86.9 | 99.9 | 98.8 |

- After separate training, the top 40 principal components explain almost all of the variance ($\geq$96.8%) in both the EEG and image encodings on their data (columns iv and viii, for rows i–iv; Table 9). This implies that the separately trained EEG and image encoders produce encodings with little more than class information on their data.

Collectively this suggests that after either joint or separate training the image encodings contain class information and mostly class information. Likewise, after separate training on their data, the EEG encodings contain primarily class information. This is not surprising, as has been noted, their data is confounded. This supports claim B and calls claims II–IV into question.

Tables 2 and 3 show the accuracy of classifying the encodings of both modalities ('EEG' and 'image') for all image encoders and datasets, 'before' joint or separate training, 'after joint' training, both with and without 'pretraining,' and 'after separate' training, using the raw encodings ('original') or the encodings reconstructed from the 'top 40' or 'bottom 960' principal components, both when considering all ILSVRC 2012 classes ('1000 classes') and when only considering the '40 classes' used by Palazzo et al. (2021).

- Image classification accuracy is near perfect with all components with 40 classes ($\geq$95.5%; column i, for rows v–viii, xiii–xvi, and xxi–xxiv; Table 10) and very high with 1000 classes ($\geq$77.9%; column xiii, for rows v–viii, xiii–xvi, and xxi–xxiv; Table 11) before training. Likewise with just the top 40 principal components ($\geq$ 85.7% and $\geq$72.2%; columns v and xvii, for rows v–viii, xiii–xvi, and xxi–xxiv; Table 12). This holds both for 'their' data and the 'block' and 'randomized' data. This demonstrates that they are giving class information implicitly to their training by use of image encoders pretrained on ImageNet, supporting claim A, and calling claim I into question.
- Image classification accuracy is much lower ($\leq$46.0%) before training when discarding the top 40 principal components (columns ix and xxi, for rows v–viii, xiii–xvi, and xxi–xxiv; Table 13). This holds both for 40 classes and 1000 classes and both for 'their' data and the 'block' and 'randomized' data. Thus there is much less class information outside the top 40 principal components. This demonstrates that they are giving little more than class information implicitly to their training by use of image encoders pretrained on ImageNet, further supporting claim A and calling claim I into question.
- With only five exceptions, all in the bottom 960 principal components, image classification accuracy decreases after joint training with pretraining (compare columns ii, vi, x, xiv, xviii, and xxii to i, v, ix, xiii, xvii, and xxi, respectively, for rows v–viii, xiii–xvi, and xxi–xxiv; Table 14). This holds both for 40 classes and 1000 classes, both for 'their' data and the

'block' and 'randomized' data, and whether using all components or just the 40 principal components. This suggests that joint training is hurting, not helping, supporting claim C(2).

- With the exception of a small number of cases that are marginally above chance, EEG classification accuracy is at chance except after separate training on confounded data either on all components or the top 40 components (columns iv, viii, xvi, and xx, for rows i–iv and ix–xii; Table 15). It is also above chance after separate training on block data in the bottom 960 components (columns xii and xxiv, for rows ix—xii; Table 16). This suggests that separate training is able to extract class from confounded data but not nonconfounded data and that joint training is not able to extract class from any data, supporting claims C(1) and D.

- While the image encoder can sometimes generalize (some of columns ii, iv, vi, viii, x, xii, xiv, xvi, xviii, xx, xxii, and xxiv, for rows v–viii, xiii–xvi, and xxi–xxiv, are above chance; Table 17) and the EEG encoder can sometimes generalize on confounded data with separate training (some of columns iv, viii, xii, xvi, xx, and xxiv, for rows i–iv and ix–xii, are above chance; Table 18), the EEG encoder cannot generalize on nonconfounded data with joint training with pretraining (all of columns ii, vi, x, xiv, xviii, and xxi, for rows xvii-xx, are at chance; Table 19). This supports claim D.

- Classification accuracy is at chance after joint training without pretraining (columns iii vii, xi, xv, xix, and xxiii; Table 20). This holds both for EEG classification and image classification, both for 40 classes and 1000 classes, both for 'their' data and the 'block' and 'randomized' data, and whether using all components, just the 40 principal components, or the bottom 960 components. This suggests that their method completely breaks down when not provided with class information through pretraining, and further supports claim A and calls claim I into question.

The results in Tables 2 and 3 exhibit some differences between different image encoders for each analysis ('EEG' *vs.* 'image,' '40 classes' *vs.* '1000 classes,' 'original' *vs.* 'top 40' *vs.* 'bottom 960,' and 'before' *vs.* 'after joint' *vs.* 'after separate'). This is not surprising. Different image classifiers exhibit different classification accuracy, especially when pretrained with different training regimens. It is further not surprising that this difference manifests with different PCA reconstructions with different components and also manifests when subjected to joint fine tuning with noisy EEG data or separate fine tuning on the tiny subset of ImageNet used here, because fine tuning can break different models in different ways. None of these differences impact our claims A–D or the refutation of claims I–IV from Palazzo et al. (2021).

Table 4 reports the per-sample triplet loss on the training set, after 'joint' training, both with and without 'pretraining,' and 'separate' training, for all image encoders and datasets.

- With only two exceptions, separate training gets to a lower loss than joint training with pretraining (compare columns vii, viii, and ix to i, ii, and iii, respectively; Table 21). This critically suggests that there is a point within the representational-capacity space of their model and loss function that achieves lower loss than achieved by their joint-training procedure. The joint-training procedure could have achieved it, it just didn't. The fact that point was achieved with separate training, indicates that the resulting EEG encodings do not have any image information and the resulting image encodings do not have any brain-activity information. This supports claim B and calls claims II–IV into question.

- With only three exceptions, joint training without pretraining gets to a lower loss than with pretraining (compare columns iv, v, and vi to i, ii, and iii, respectively; Table 22). Yet classification accuracy is at chance after joint training without pretraining (columns iii vii, xi, xv, xix, and xxiii of Tables 2 and 3; Table 20). This suggests that joint training without pretraining can memorize the training set yet fail to generalize at all, and further supports claim A and calls claim I into question.

Table 5 reports the per-sample MSE loss of the individual encoders of both modalities ('EEG' and 'image') on the training set, after separate training, for all image encoders and datasets.

- Separate training against one-hot labels can get to very low MSE (all columns and rows), indicating that the model has the representational capacity to memorize both the EEG and image training data. Achieving this point in the representational-capacity space with separate training and the encodings being one-hot implies that at this point, the EEG and image encodings have nothing but class information. Since this point could have been achieved

Table 2: Original classification accuracy and classification accuracy after reconstruction (%), using just the encodings, 'before' joint or separate training, 'after joint' training, both with and without 'pretraining,' and 'after separate' training. Rows i–vii 'their,' rows ix–xvi 'block,' and rows xvii–xxiv 'randomized'. Rows i–iv, ix–xii, and xvii–xx EEG. Rows v–vii, xiii–xvi, and xxi–xxiv Image. Rows i, v, ix, xiii, xvii, and xxi Inception v2. Rows ii, vi, x, xiv, xviii, and xxii ResNet-101. Rows iii, vii, xi, xv, xix, and xxiii DenseNet-161. Rows iv, viii, xii, xvi, xx, and xxiv AlexNet.

| | original | | | | 40 classes top 40 | | | | bottom 960 | | | |
| | before | after joint | | after separate | before | after joint | | after separate | before | after joint | | after separate |
| | | pretraining | no pretraining | | | pretraining | no pretraining | | | pretraining | no pretraining | |
| | i | ii | iii | iv | v | vi | vii | viii | ix | x | xi | xii |
| --- | --- | --- | --- | --- | --- | --- | --- | --- | --- | --- | --- | --- |
| i | 2.6 | 3.0 | 2.5 | 55.5 | 2.2 | 2.6 | 2.5 | 55.5 | 2.5 | 3.1 | 2.5 | 2.3 |
| ii | 2.3 | 3.0 | 2.6 | 54.7 | 2.8 | 2.4 | 2.4 | 54.7 | 2.4 | 3.1 | 2.5 | 2.3 |
| iii | 2.5 | 3.1 | 2.4 | 57.4 | 2.4 | 2.6 | 2.3 | 57.4 | 2.5 | 3.2 | 2.4 | 2.5 |
| iv | 2.3 | 2.5 | 2.4 | 57.3 | 2.4 | 2.6 | 2.6 | 57.3 | 2.4 | 2.5 | 2.6 | 2.5 |
| v | 99.7 | 43.2 | 2.7 | 3.4 | 99.4 | 13.5 | 2.9 | 2.2 | 45.5 | 29.4 | 2.7 | 2.5 |
| vi | 99.4 | 24.2 | 2.4 | 82.5 | 98.3 | 7.3 | 2.9 | 70.5 | 33.0 | 9.5 | 2.1 | 39.4 |
| vii | 99.2 | 7.2 | 2.4 | 23.3 | 97.9 | 4.4 | 2.3 | 3.5 | 42.7 | 3.8 | 2.6 | 2.6 |
| viii | 95.8 | 2.6 | 2.7 | 18.7 | 85.8 | 2.6 | 2.7 | 19.1 | 34.1 | 2.6 | 2.4 | 2.4 |
| ix | 2.4 | 6.8 | 1.9 | 77.0 | 1.8 | 7.1 | 2.0 | 69.8 | 2.6 | 3.6 | 2.5 | 32.1 |
| x | 1.6 | 3.5 | 2.2 | 76.6 | 1.1 | 4.2 | 2.5 | 70.5 | 2.4 | 2.8 | 2.2 | 28.1 |
| xi | 1.4 | 4.1 | 2.0 | 77.0 | 1.2 | 3.8 | 1.9 | 72.2 | 2.5 | 3.5 | 2.7 | 25.9 |
| xii | 2.9 | 4.1 | 2.9 | 76.5 | 2.5 | 4.2 | 2.2 | 70.0 | 2.0 | 3.4 | 2.5 | 27.8 |
| xiii | 99.8 | 96.4 | 3.0 | 9.3 | 99.4 | 94.7 | 3.1 | 2.9 | 46.0 | 46.8 | 2.4 | 2.5 |
| xiv | 99.3 | 75.0 | 2.4 | 11.0 | 98.5 | 68.5 | 2.7 | 3.5 | 34.1 | 14.7 | 2.5 | 3.7 |
| xv | 99.1 | 95.8 | 2.5 | 5.7 | 97.7 | 86.8 | 2.9 | 3.8 | 43.9 | 50.9 | 2.1 | 2.6 |
| xvi | 95.5 | 60.9 | 2.5 | 3.5 | 85.7 | 50.9 | 2.4 | 3.2 | 34.6 | 15.6 | 2.5 | 2.3 |
| xvii | 2.6 | 2.5 | 2.5 | 1.6 | 2.6 | 2.9 | 2.6 | 2.5 | 3.0 | 2.2 | 2.7 | 1.6 |
| xviii | 2.6 | 2.4 | 2.8 | 2.1 | 2.8 | 2.7 | 2.9 | 2.3 | 2.6 | 2.7 | 3.0 | 1.8 |
| xix | 2.1 | 2.2 | 2.3 | 1.5 | 2.5 | 2.3 | 2.5 | 2.0 | 3.1 | 2.0 | 1.9 | 1.9 |
| xx | 2.5 | 2.8 | 3.0 | 2.2 | 2.2 | 2.4 | 2.9 | 1.6 | 2.6 | 2.8 | 3.1 | 2.4 |
| xxi | 99.8 | 91.1 | 3.4 | 9.4 | 99.4 | 84.7 | 3.0 | 2.9 | 46.0 | 57.1 | 2.1 | 2.5 |
| xxii | 99.3 | 69.8 | 2.9 | 11.2 | 98.5 | 53.1 | 2.6 | 4.0 | 34.1 | 9.4 | 3.1 | 2.2 |
| xxiii | 99.1 | 65.6 | 2.1 | 5.1 | 97.7 | 36.0 | 2.5 | 3.0 | 43.9 | 36.1 | 2.3 | 3.0 |
| xxiv | 95.5 | 3.6 | 2.4 | 3.5 | 85.7 | 3.6 | 2.5 | 3.5 | 34.6 | 2.5 | 2.5 | 2.5 |

Table 3: Original classification accuracy and classification accuracy after reconstruction (%), using just the encodings, 'before' joint or separate training, 'after joint' training, both with and without 'pretraining,' and 'after separate' training. Rows i–vii 'their,' rows ix–xvi 'block,' and rows xvii–xxiv 'randomized'. Rows i–iv, ix–xii, and xvii–xx EEG. Rows v–vii, xiii–xvi, and xxi–xxiv Image. Rows i, v, ix, xiii, xvii, and xxi Inception v2. Rows ii, vi, x, xiv, xviii, and xxii ResNet-101. Rows iii, vii, xi, xv, xix, and xxiii DenseNet-161. Rows iv, viii, xii, xvi, xx, and xxiv AlexNet.

| | original | | | | 1000 classes top 40 | | | | bottom 960 | | | |
| | before | after joint | | after separate | before | after joint | | after separate | before | after joint | | after separate |
| | | pretraining | no pretraining | | | pretraining | no pretraining | | | pretraining | no pretraining | |
| | xiii | xiv | xv | xvi | xvii | xviii | xix | xx | xxi | xxii | xxiii | xxiv |
|---|---|---|---|---|---|---|---|---|---|---|---|---|
| i | 0.1 | 0.1 | 0.1 | 55.5 | 0.1 | 0.1 | 0.1 | 55.5 | 0.1 | 0.1 | 0.1 | 2.3 |
| ii | 0.0 | 0.1 | 0.1 | 54.7 | 0.0 | 0.0 | 0.1 | 54.7 | 0.1 | 0.1 | 0.1 | 2.3 |
| iii | 0.1 | 0.1 | 0.1 | 57.4 | 0.0 | 0.1 | 0.0 | 57.4 | 0.1 | 0.1 | 0.1 | 2.5 |
| iv | 0.1 | 0.1 | 0.1 | 57.3 | 0.0 | 0.1 | 0.1 | 57.3 | 0.1 | 0.1 | 0.1 | 2.5 |
| v | 92.8 | 16.5 | 0.1 | 0.0 | 98.9 | 4.0 | 0.0 | 0.0 | 12.8 | 9.3 | 0.0 | 0.0 |
| vi | 90.5 | 2.6 | 0.0 | 82.5 | 95.6 | 0.3 | 0.0 | 70.5 | 9.5 | 0.7 | 0.0 | 39.2 |
| vii | 89.4 | 0.1 | 0.0 | 12.1 | 95.8 | 0.0 | 0.0 | 0.1 | 14.2 | 0.0 | 0.1 | 0.0 |
| viii | 78.4 | 0.0 | 0.3 | 18.7 | 72.5 | 0.0 | 0.3 | 19.1 | 12.5 | 0.0 | 0.0 | 2.4 |
| ix | 0.1 | 0.4 | 0.0 | 77.0 | 0.0 | 0.2 | 0.1 | 69.8 | 0.0 | 0.2 | 0.1 | 28.0 |
| x | 0.0 | 0.1 | 0.1 | 76.6 | 0.0 | 0.1 | 0.1 | 70.5 | 0.0 | 0.0 | 0.1 | 24.0 |
| xi | 0.1 | 0.1 | 0.0 | 77.0 | 0.0 | 0.0 | 0.0 | 72.2 | 0.1 | 0.1 | 0.1 | 22.3 |
| xii | 0.0 | 0.1 | 0.1 | 76.4 | 0.0 | 0.3 | 0.2 | 70.0 | 0.0 | 0.0 | 0.1 | 24.2 |
| xiii | 93.0 | 75.1 | 0.1 | 0.5 | 98.8 | 87.0 | 0.0 | 0.0 | 13.2 | 11.7 | 0.0 | 0.0 |
| xiv | 90.5 | 56.4 | 0.0 | 2.0 | 96.2 | 52.4 | 0.0 | 0.1 | 9.8 | 0.2 | 0.0 | 0.0 |
| xv | 89.5 | 71.1 | 0.1 | 0.2 | 96.4 | 67.2 | 0.4 | 0.2 | 15.0 | 17.9 | 0.0 | 0.0 |
| xvi | 77.9 | 25.4 | 0.0 | 2.6 | 72.2 | 19.7 | 0.0 | 2.5 | 12.9 | 4.2 | 0.0 | 2.3 |
| xvii | 0.1 | 0.1 | 0.1 | 1.5 | 0.1 | 0.1 | 0.0 | 2.5 | 0.2 | 0.1 | 0.2 | 1.4 |
| xviii | 0.0 | 0.1 | 0.1 | 1.9 | 0.0 | 0.1 | 0.2 | 2.3 | 0.1 | 0.2 | 0.1 | 1.6 |
| xix | 0.1 | 0.1 | 0.1 | 1.5 | 0.0 | 0.0 | 0.1 | 2.0 | 0.1 | 0.1 | 0.1 | 1.7 |
| xx | 0.2 | 0.1 | 0.2 | 2.2 | 0.3 | 0.0 | 0.1 | 1.6 | 0.1 | 0.1 | 0.1 | 1.9 |
| xxi | 93.0 | 64.0 | 0.1 | 0.6 | 98.8 | 73.3 | 0.0 | 0.0 | 13.2 | 20.4 | 0.0 | 0.0 |
| xxii | 90.5 | 42.0 | 0.0 | 2.0 | 96.2 | 28.5 | 0.0 | 0.1 | 9.8 | 3.5 | 0.0 | 0.0 |
| xxiii | 89.5 | 30.5 | 0.1 | 0.0 | 96.4 | 11.7 | 0.0 | 0.0 | 15.0 | 13.6 | 0.0 | 0.0 |
| xxiv | 77.9 | 0.0 | 0.0 | 2.6 | 72.2 | 0.0 | 0.0 | 2.6 | 12.9 | 0.0 | 0.0 | 2.5 |

Table 4: Per-sample triplet loss on the training set, after joint training, both with and without 'pretraining,' and after separate training, averaged over samples and splits.

| | | joint | | | | | | separate | | |
| | | pretraining | | | no pretraining | | | | | |
| | | their | block | randomized | their | block | randomized | their | block | randomized |
| | | i | ii | iii | iv | v | vi | vii | viii | ix |
| Inception v3 | i | 1.303 | 1.098 | 0.932 | 1.074 | 0.938 | 0.919 | 1.150 | 1.027 | 1.104 |
| ResNet-101 | ii | 1.155 | 0.925 | 1.019 | 0.963 | 1.006 | 1.080 | 0.885 | 0.936 | 0.884 |
| DenseNet-161 | iii | 1.027 | 1.080 | 1.047 | 0.988 | 0.919 | 0.929 | 1.007 | 0.982 | 1.029 |
| AlexNet | iv | 1.417 | 1.142 | 0.981 | 1.136 | 1.055 | 1.034 | 0.925 | 0.956 | 0.935 |

Table 5: Per-sample MSE loss of the individual encoders on the training set, after separate training, averaged over samples and splits.

| | | EEG | | | image | | |
| | | their | block | randomized | their | block | randomized |
| | | i | ii | iii | iv | v | vi |
| Inception v3 | i | 0.001 | 0.001 | 0.001 | 0.002 | 0.010 | 0.010 |
| ResNet-101 | ii | 0.001 | 0.001 | 0.001 | 0.001 | 0.002 | 0.002 |
| DenseNet-161 | iii | 0.001 | 0.001 | 0.001 | 0.001 | 0.002 | 0.002 |
| AlexNet | iv | 0.001 | 0.001 | 0.001 | 0.001 | 0.001 | 0.001 |

with joint training, joint training could have produced a set of encodings that have nothing but class information. This supports claim B and calls claims II–IV into question.

Collectively our results[3] suggest that:

- Their statement (claim I) about not incorporating any class information in joint training is false (our claim A).
- Their encoder models combined with their triplet loss function allow a point in the space where the encodings on the training set are one-hot and thus cannot possibly contain anything other than class information. This point has zero triplet loss and thus is the minimum. It is achievable. Thus their framework can produce encodings with nothing but class information.
- This point can be reached by an alternate separate-training regimen that clearly does not transfer any information between the EEG and image encoders.

It is highly unlikely that the suboptimal EEG model that they produce by joint training has any image information, the suboptimal image model that they produce by joint training has any brain-activity information, and neither suboptimal model produced by joint training has anything other than class information, even when run on nonconfounded data. This supports claim B and calls claims II–IV into question.

## 4 CONCLUSION

Several independent lines of research have completely refuted a large body of completely flawed work (Spampinato et al., 2017; Palazzo et al., 2017; Kavasidis et al., 2017; Tirupattur et al., 2018; Palazzo et al., 2020; 2021) along completely different axes. Li et al. (2021) demonstrated that the dataset used, and the methods used to collect that dataset, suffer from a temporal confound, correlating stimulus class with experiment timing. Accuracy drops to chance when the confound is removed. Ahmed et al. (2021) demonstrated that this holds even with a much larger dataset. Ahmed et al. (2022) demonstrated that this holds for the additional classifiers used in Palazzo et al. (2020; 2021). Bharadwaj et al. (2023) demonstrated that this holds even when using supertrials. This is important because this invalidates a huge body of work that uses that dataset and other datasets collected with the same methods. See Ahmed et al. (2021).

---

[3]The raw data that produced these results is available at http://dx.doi.org/10.21227/x2gf-5324. Our code, included in the supplementary material, is built on top of the code in http://dx.doi.org/10.21227/x2gf-5324.

Here we show that not only are the dataset and data collection methods fundamentally flawed, the joint training regimen is also fundamentally flawed. This is important not only because the confounded dataset and flawed collection methods continue to be used, but also because the same joint training regimen continues to be used (Bai et al., 2023; Zeng et al., 2023; Ahmadieh et al., 2023; Lan et al., 2023; Du et al., 2023; Liu et al., 2023; Song et al., 2023; Ye et al., 2024). None of the work that jointly trains EEG and image classifiers even attempts to assess whether joint training is effective in information transfer. In most cases, we can't perform the analysis because the published work does not contain sufficient information to do so, including but not limited to stimulus timing and presentation order. Without this, one cannot even be sure that the data is free from confounds.

We implore all future EEG image classification effort to release **raw data** that includes stimulus timing and presentation order, not preprocessed data. We further implore all future effort to jointly train EEG and image classifiers to employ the methods presented here to assess the effectiveness of purported information transfer attributed to joint training, and not to assume such effectiveness due to classification accuracy and image reconstruction.

Recent progress in computer vision and machine learning has been gauged largely by improvement in performance metrics of methods on datasets and results that 'look good.' Palazzo et al. (2021) largely base their claims on similar criteria. While synthetic engineering methods can be evaluated by the utility of artifacts, analytic scientific claims about the underlying physical world can only be evaluated by finding hypotheses that resist falsification. We have falsified all of the claims of Palazzo et al. (2021).

## AUTHOR CONTRIBUTIONS

REDACTED

## ACKNOWLEDGMENTS

REDACTED

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

# A  METHOD

We report all results for three sets of EEG data: the data reported by Spampinato et al. (2017) as used by Palazzo et al. (2021) ('their'), the data reported by Li et al. (2021) for subject 6 first image block run ('block'), and the data reported by Li et al. (2021) for subject 6 image rapid-event run ('randomized'). We report all results for four off-the-shelf image classifiers taken as image encoders: Inception v3 (Szegedy et al., 2016), ResNet-101 (He et al., 2016), DenseNet-161 (Huang et al., 2017), and AlexNet (Krizhevsky et al., 2012). We report all results both before joint training ('before'), where the EEG encoder is randomly initialized, after joint training ('after joint'), and after separate training ('after separate'). For 'before' and 'after separate' the image encoder is initialized with off-the-shelf weights trained on the ILSVRC 2012 training set (Russakovsky et al., 2014). For 'joint', we report results both where the image encoder is initialized with off-the-shelf weights trained on the ILSVRC 2012 training set (Russakovsky et al., 2014) ('pretraining') and where the image encoder is randomly initialized ('no pretraining'). The EEG and image encoders both output a 1000 element encoding. During joint training, a triplet loss is used to drive the EEG and image encodings to have a high dot product when the EEG signal is associated with the stimulus image that elicited the associated EEG signal and a low dot product when the EEG signal is associated with a different image, irrespective of class. We do not postpend a classifier (ReLU, $1000 \rightarrow 40$ FC, softmax) to the encoders, either during training or after training. All analyses are done directly on the unmodified encodings produced by the encoders. For both the EEG encodings ('EEG') and the image encodings ('image'), we analyze the unmodified encodings ('original') and encodings reconstructed by principal component analysis ('PCA'). For PCA, we analyze two reconstructions: one with only the top 40 principal components ('top 40') and one with only the bottom 960 principal components ('bottom 960'). We classify original and reconstructed EEG and image encodings, before and after joint and separate training, in two ways. In the first, we simply select the label of the maximal element among all 1000 elements ('1000 classes'). In the second, we select the label of the maximal element among only the 40 classes used as stimuli ('40 classes'). In both cases, correct labels are considered positives. In the former, negatives can result from incorrect labels, both within and outside the subset of 40 ILSVRC classes used as stimuli. In the latter, negatives can result only from incorrect labels within the subset of 40 ILSVRC classes used as stimuli.

Table 6: The top 40 principal components explain a large portion of the image-encoding variance (≥63.9%) before training (column v). This together with Tables 2 and 3 implies that the pretrained image encoders produce encodings that contain (primarily) class information on all data. This supports claim A and calls claim I into question.

| | | | EEG | | | | image | | | |
| | | | before | after joint | | after separate | before | after joint | | after separate |
| | | | | pretraining | no pretraining | | | pretraining | no pretraining | |
| | | | i | ii | iii | iv | v | vi | vii | viii |
| their | Inception v3 | i | 44.7 | 37.2 | 15.3 | 99.5 | 64.1 | 94.9 | 98.9 | 100.0 |
| | ResNet-101 | ii | 46.5 | 36.2 | 17.7 | 99.5 | 82.0 | 98.9 | 98.2 | 96.8 |
| | DenseNet-161 | iii | 48.2 | 25.6 | 22.7 | 100.0 | 75.4 | 97.4 | 90.2 | 100.0 |
| | AlexNet | iv | 46.3 | 21.7 | 18.5 | 99.5 | 82.3 | nan | 100.0 | 99.7 |
| block | Inception v3 | v | 83.3 | 75.9 | 69.4 | 42.3 | 63.9 | 57.4 | 98.3 | 99.9 |
| | ResNet-101 | vi | 85.4 | 79.8 | 72.8 | 41.9 | 81.9 | 99.5 | 99.0 | 99.7 |
| | DenseNet-161 | vii | 84.0 | 79.3 | 76.3 | 42.9 | 75.3 | 69.4 | 94.3 | 99.5 |
| | AlexNet | viii | 84.4 | 78.9 | 73.2 | 40.0 | 82.2 | 78.9 | 99.9 | 98.9 |
| randomized | Inception v3 | ix | 62.1 | 40.5 | 38.3 | 19.3 | 63.9 | 60.2 | 98.1 | 99.9 |
| | ResNet-101 | x | 61.9 | 52.2 | 43.6 | 19.7 | 81.9 | 99.1 | 98.8 | 99.8 |
| | DenseNet-161 | xi | 63.5 | 48.5 | 46.5 | 19.6 | 75.3 | 70.8 | 93.9 | 99.7 |
| | AlexNet | xii | 63.6 | 49.5 | 41.3 | 19.2 | 82.2 | 86.9 | 99.9 | 98.8 |

Table 7: After joint training with pretraining on their data, the top 40 principal components explain almost all (≥94.9%) of the variance in the image encodings (column vi, for rows i–iv). This implies that the image encoders jointly trained on their data produce encodings with little more than class information.

| | | | EEG | | | | image | | | |
| | | | before | after joint | | after separate | before | after joint | | after separate |
| | | | | pretraining | no pretraining | | | pretraining | no pretraining | |
| | | | i | ii | iii | iv | v | vi | vii | viii |
| their | Inception v3 | i | 44.7 | 37.2 | 15.3 | 99.5 | 64.1 | 94.9 | 98.9 | 100.0 |
| | ResNet-101 | ii | 46.5 | 36.2 | 17.7 | 99.5 | 82.0 | 98.9 | 98.2 | 96.8 |
| | DenseNet-161 | iii | 48.2 | 25.6 | 22.7 | 100.0 | 75.4 | 97.4 | 90.2 | 100.0 |
| | AlexNet | iv | 46.3 | 21.7 | 18.5 | 99.5 | 82.3 | nan | 100.0 | 99.7 |
| block | Inception v3 | v | 83.3 | 75.9 | 69.4 | 42.3 | 63.9 | 57.4 | 98.3 | 99.9 |
| | ResNet-101 | vi | 85.4 | 79.8 | 72.8 | 41.9 | 81.9 | 99.5 | 99.0 | 99.7 |
| | DenseNet-161 | vii | 84.0 | 79.3 | 76.3 | 42.9 | 75.3 | 69.4 | 94.3 | 99.5 |
| | AlexNet | viii | 84.4 | 78.9 | 73.2 | 40.0 | 82.2 | 78.9 | 99.9 | 98.9 |
| randomized | Inception v3 | ix | 62.1 | 40.5 | 38.3 | 19.3 | 63.9 | 60.2 | 98.1 | 99.9 |
| | ResNet-101 | x | 61.9 | 52.2 | 43.6 | 19.7 | 81.9 | 99.1 | 98.8 | 99.8 |
| | DenseNet-161 | xi | 63.5 | 48.5 | 46.5 | 19.6 | 75.3 | 70.8 | 93.9 | 99.7 |
| | AlexNet | xii | 63.6 | 49.5 | 41.3 | 19.2 | 82.2 | 86.9 | 99.9 | 98.8 |

Table 8: After separate training, the top 40 principal components explain almost all of the variance (≥96.8%) in the image encodings (column viii). This implies that the separately trained image encoders produce encodings with little more than class information on all data.

| | | | EEG | | | | image | | | |
| | | | before | after joint | | after separate | before | after joint | | after separate |
| | | | | pretraining | no pretraining | | | pretraining | no pretraining | |
| | | | i | ii | iii | iv | v | vi | vii | viii |
| their | Inception v3 | i | 44.7 | 37.2 | 15.3 | 99.5 | 64.1 | 94.9 | 98.9 | 100.0 |
| | ResNet-101 | ii | 46.5 | 36.2 | 17.7 | 99.5 | 82.0 | 98.9 | 98.2 | 96.8 |
| | DenseNet-161 | iii | 48.2 | 25.6 | 22.7 | 100.0 | 75.4 | 97.4 | 90.2 | 100.0 |
| | AlexNet | iv | 46.3 | 21.7 | 18.5 | 99.5 | 82.3 | nan | 100.0 | 99.7 |
| block | Inception v3 | v | 83.3 | 75.9 | 69.4 | 42.3 | 63.9 | 57.4 | 98.3 | 99.9 |
| | ResNet-101 | vi | 85.4 | 79.8 | 72.8 | 41.9 | 81.9 | 99.5 | 99.0 | 99.7 |
| | DenseNet-161 | vii | 84.0 | 79.3 | 76.3 | 42.9 | 75.3 | 69.4 | 94.3 | 99.5 |
| | AlexNet | viii | 84.4 | 78.9 | 73.2 | 40.0 | 82.2 | 78.9 | 99.9 | 98.9 |
| randomized | Inception v3 | ix | 62.1 | 40.5 | 38.3 | 19.3 | 63.9 | 60.2 | 98.1 | 99.9 |
| | ResNet-101 | x | 61.9 | 52.2 | 43.6 | 19.7 | 81.9 | 99.1 | 98.8 | 99.8 |
| | DenseNet-161 | xi | 63.5 | 48.5 | 46.5 | 19.6 | 75.3 | 70.8 | 93.9 | 99.7 |
| | AlexNet | xii | 63.6 | 49.5 | 41.3 | 19.2 | 82.2 | 86.9 | 99.9 | 98.8 |

Table 9: After separate training, the top 40 principal components explain almost all of the variance ($\geq$96.8%) in both the EEG and image encodings on their data (columns iv and viii, for rows i–iv). This implies that the separately trained EEG and image encoders produce encodings with little more than class information on their data.

| | | | EEG | | | | image | | | |
| | | | before | after joint | | after separate | before | after joint | | after separate |
| | | | | pretraining | no pretraining | | | pretraining | no pretraining | |
| | | | i | ii | iii | iv | v | vi | vii | viii |
|---|---|---|---|---|---|---|---|---|---|---|
| their | Inception v3 | i | 44.7 | 37.2 | 15.3 | 99.5 | 64.1 | 94.9 | 98.9 | 100.0 |
| | ResNet-101 | ii | 46.5 | 36.2 | 17.7 | 99.5 | 82.0 | 98.9 | 98.2 | 96.8 |
| | DenseNet-161 | iii | 48.2 | 25.6 | 22.7 | 100.0 | 75.4 | 97.4 | 90.2 | 100.0 |
| | AlexNet | iv | 46.3 | 21.7 | 18.5 | 99.5 | 82.3 | nan | 100.0 | 99.7 |
| block | Inception v3 | v | 83.3 | 75.9 | 69.4 | 42.3 | 63.9 | 57.4 | 98.3 | 99.9 |
| | ResNet-101 | vi | 85.4 | 79.8 | 72.8 | 41.9 | 81.9 | 99.5 | 99.0 | 99.7 |
| | DenseNet-161 | vii | 84.0 | 79.3 | 76.3 | 42.9 | 75.3 | 69.4 | 94.3 | 99.5 |
| | AlexNet | viii | 84.4 | 78.9 | 73.2 | 40.0 | 82.2 | 78.9 | 99.9 | 98.9 |
| randomized | Inception v3 | ix | 62.1 | 40.5 | 38.3 | 19.3 | 63.9 | 60.2 | 98.1 | 99.9 |
| | ResNet-101 | x | 61.9 | 52.2 | 43.6 | 19.7 | 81.9 | 99.1 | 98.8 | 99.8 |
| | DenseNet-161 | xi | 63.5 | 48.5 | 46.5 | 19.6 | 75.3 | 70.8 | 93.9 | 99.7 |
| | AlexNet | xii | 63.6 | 49.5 | 41.3 | 19.2 | 82.2 | 86.9 | 99.9 | 98.8 |

Table 10: Image classification accuracy is near perfect with all components with 40 classes (≥95.5%; column i, for rows v–viii, xiii–xvi, and xxi–xxiv) before training. This holds both for 'their' data, 'block' data, and 'randomized' data. This, together with Tables 11 and 12, demonstrates that they are giving class information implicitly to their training by use of image encoders pretrained on ImageNet, supports claim A, and calls claim I into question.

| group | mod. | model | | original (40 classes) | | | | top-40 | | | | bottom 960 | | | | original (1000 classes) | | | | top-40 | | | | bottom 960 | | |
|---|---|---|---|---|---|---|---|---|---|---|---|---|---|---|---|---|---|---|---|---|---|---|---|---|---|---|---|
| | | | before | joint pre | joint no | sep | before | joint pre | joint no | sep | before | joint pre | joint no | sep | before | joint pre | joint no | sep | before | joint pre | joint no | sep | before | joint pre | joint no | sep |
| | | | i | ii | iii | iv | v | vi | vii | viii | ix | x | xi | xii | xiii | xiv | xv | xvi | xvii | xviii | xix | xx | xxi | xxii | xxiii | xxiv |
| their | EEG | Inception v3 | 2.6 | 3.0 | 2.5 | 55.5 | 2.2 | 2.6 | 2.5 | 55.5 | 2.5 | 3.1 | 2.5 | 2.3 | 0.1 | 0.1 | 0.1 | 55.5 | 0.1 | 0.0 | 0.1 | 55.5 | 0.1 | 0.1 | 0.1 | 2.3 |
| | | ResNet-101 | 2.3 | 3.0 | 2.6 | 54.7 | 2.8 | 2.4 | 2.4 | 54.7 | 2.4 | 3.1 | 2.5 | 2.5 | 0.0 | 0.1 | 0.1 | 54.7 | 0.0 | 0.0 | 0.0 | 54.7 | 0.1 | 0.1 | 0.1 | 2.3 |
| | | DenseNet-161 | 2.5 | 3.1 | 2.4 | 57.4 | 2.4 | 2.6 | 2.3 | 57.4 | 2.5 | 3.2 | 2.4 | 2.5 | 0.1 | 0.1 | 0.1 | 57.4 | 0.0 | 0.1 | 0.0 | 57.4 | 0.1 | 0.1 | 0.1 | 2.5 |
| | | AlexNet | 2.3 | 2.5 | 2.4 | 57.3 | 2.4 | 2.6 | 2.6 | 57.3 | 2.4 | 2.5 | 2.6 | 2.5 | 0.1 | 0.1 | 0.1 | 57.3 | 0.0 | 0.1 | 0.1 | 57.3 | 0.1 | 0.1 | 0.1 | 2.5 |
| | image | Inception v3 | 99.7 | 43.2 | 2.7 | 3.4 | 99.4 | 13.5 | 2.9 | 2.2 | 45.5 | 29.4 | 2.7 | 2.5 | 92.8 | 16.5 | 0.0 | 0.0 | 98.9 | 4.0 | 0.0 | 0.0 | 12.8 | 9.3 | 0.0 | 0.0 |
| | | ResNet-101 | 99.4 | 24.2 | 2.4 | 82.5 | 98.3 | 7.3 | 2.9 | 70.5 | 33.0 | 9.5 | 2.1 | 39.4 | 90.5 | 2.6 | 0.0 | 82.5 | 95.6 | 0.3 | 0.0 | 70.5 | 9.5 | 0.7 | 0.0 | 39.2 |
| | | DenseNet-161 | 99.2 | 7.2 | 2.4 | 23.3 | 97.9 | 4.4 | 2.3 | 3.5 | 42.7 | 3.8 | 2.6 | 2.6 | 89.4 | 0.1 | 0.0 | 12.1 | 95.8 | 0.0 | 0.0 | 0.1 | 14.2 | 0.1 | 0.1 | 0.0 |
| | | AlexNet | 95.8 | 2.6 | 2.7 | 18.7 | 85.8 | 2.6 | 2.7 | 19.1 | 34.1 | 2.6 | 2.4 | 2.4 | 78.4 | 0.0 | 0.3 | 18.7 | 72.5 | 0.0 | 0.3 | 19.1 | 12.5 | 0.0 | 0.0 | 2.4 |
| block | EEG | Inception v3 | 2.4 | 6.8 | 2.2 | 77.0 | 1.8 | 7.1 | 2.0 | 69.8 | 2.6 | 3.6 | 2.5 | 32.1 | 0.1 | 0.4 | 0.0 | 77.0 | 0.0 | 0.2 | 0.1 | 69.8 | 0.0 | 0.2 | 0.1 | 28.0 |
| | | ResNet-101 | 1.6 | 3.5 | 2.2 | 76.6 | 1.1 | 4.2 | 2.5 | 70.5 | 2.4 | 2.8 | 2.2 | 28.1 | 0.0 | 0.1 | 0.1 | 76.6 | 0.0 | 0.1 | 0.1 | 70.5 | 0.0 | 0.1 | 0.1 | 24.0 |
| | | DenseNet-161 | 1.4 | 4.1 | 2.0 | 77.0 | 1.2 | 3.8 | 1.9 | 72.2 | 2.5 | 3.5 | 2.7 | 25.9 | 0.1 | 0.1 | 0.0 | 77.0 | 0.1 | 0.0 | 0.0 | 72.2 | 0.0 | 0.1 | 0.1 | 22.3 |
| | | AlexNet | 2.9 | 4.1 | 2.9 | 76.5 | 2.5 | 4.2 | 2.2 | 70.0 | 2.0 | 3.4 | 2.5 | 27.8 | 0.0 | 0.1 | 0.1 | 76.4 | 0.0 | 0.3 | 0.2 | 70.0 | 0.0 | 0.1 | 0.1 | 24.2 |
| | image | Inception v3 | 99.8 | 96.4 | 3.0 | 9.3 | 99.4 | 94.7 | 3.1 | 2.9 | 46.0 | 46.8 | 2.4 | 2.5 | 93.0 | 75.1 | 0.1 | 0.5 | 98.8 | 87.0 | 0.0 | 0.0 | 13.2 | 1.7 | 0.0 | 0.0 |
| | | ResNet-101 | 99.3 | 75.0 | 2.4 | 11.0 | 98.5 | 68.5 | 2.7 | 3.5 | 34.1 | 14.7 | 2.5 | 3.7 | 90.5 | 56.4 | 0.1 | 2.0 | 96.2 | 52.4 | 0.0 | 0.1 | 9.8 | 0.2 | 0.0 | 0.0 |
| | | DenseNet-161 | 99.1 | 95.8 | 2.5 | 5.7 | 97.7 | 86.8 | 2.9 | 3.8 | 43.9 | 50.9 | 2.1 | 2.6 | 89.5 | 71.1 | 0.1 | 0.2 | 96.4 | 67.2 | 0.4 | 0.2 | 15.0 | 17.9 | 0.0 | 0.0 |
| | | AlexNet | 95.5 | 60.9 | 2.5 | 3.5 | 85.7 | 50.9 | 2.4 | 3.2 | 34.6 | 15.6 | 2.5 | 2.3 | 77.9 | 25.4 | 0.0 | 0.2 | 72.2 | 19.7 | 0.0 | 2.5 | 12.9 | 4.2 | 0.0 | 2.3 |
| randomized | EEG | Inception v3 | 2.6 | 2.5 | 2.5 | 1.6 | 2.6 | 2.9 | 2.6 | 2.5 | 3.0 | 2.2 | 2.7 | 1.6 | 0.1 | 0.1 | 0.1 | 1.5 | 0.1 | 0.1 | 0.2 | 2.5 | 0.2 | 0.1 | 0.2 | 1.4 |
| | | ResNet-101 | 2.6 | 2.4 | 2.8 | 2.1 | 2.8 | 2.7 | 2.9 | 2.3 | 2.6 | 2.7 | 3.0 | 1.8 | 0.0 | 0.1 | 0.1 | 1.9 | 0.0 | 0.0 | 0.1 | 2.3 | 0.1 | 0.2 | 0.1 | 1.6 |
| | | DenseNet-161 | 2.1 | 2.2 | 2.3 | 1.5 | 2.5 | 2.3 | 2.5 | 2.0 | 3.1 | 2.0 | 1.9 | 1.9 | 0.1 | 0.1 | 0.1 | 1.5 | 0.0 | 0.0 | 0.1 | 2.0 | 0.1 | 0.1 | 0.1 | 1.7 |
| | | AlexNet | 2.5 | 2.8 | 3.0 | 2.2 | 2.2 | 2.4 | 2.9 | 1.6 | 2.6 | 2.8 | 3.1 | 2.4 | 0.2 | 0.1 | 0.2 | 2.2 | 0.3 | 0.0 | 0.1 | 1.6 | 0.1 | 0.1 | 0.1 | 1.9 |
| | image | Inception v3 | 99.8 | 91.1 | 3.4 | 9.4 | 99.4 | 84.7 | 3.0 | 2.9 | 46.0 | 57.1 | 2.1 | 2.5 | 93.0 | 64.0 | 0.1 | 0.6 | 98.8 | 73.3 | 0.0 | 0.0 | 13.2 | 20.4 | 0.0 | 0.0 |
| | | ResNet-101 | 99.3 | 69.8 | 2.9 | 11.2 | 98.5 | 53.1 | 2.6 | 4.0 | 34.1 | 9.4 | 3.1 | 2.2 | 90.5 | 42.0 | 0.0 | 2.0 | 96.2 | 28.5 | 0.0 | 0.1 | 9.8 | 3.5 | 0.0 | 0.0 |
| | | DenseNet-161 | 99.1 | 65.6 | 2.1 | 5.1 | 97.7 | 36.0 | 2.5 | 3.0 | 43.9 | 36.1 | 2.3 | 3.0 | 89.5 | 30.5 | 0.1 | 0.0 | 96.4 | 11.7 | 0.0 | 0.0 | 15.0 | 13.6 | 0.0 | 0.0 |
| | | AlexNet | 95.5 | 3.6 | 2.4 | 3.5 | 85.7 | 3.6 | 2.5 | 3.5 | 34.6 | 2.5 | 2.5 | 2.5 | 77.9 | 0.0 | 0.0 | 2.6 | 72.2 | 0.0 | 0.0 | 2.6 | 12.9 | 0.0 | 0.0 | 2.5 |

Table 11: Image classification accuracy is very high with all components with 1000 classes (≥77.9%; column xiii, for rows v–viii, xiii–xvi, and xxi–xxiv) before training. This holds both for 'their' data, 'block' data, and 'randomized' data. This, together with Tables 10 and 12, demonstrates that they are giving class information implicitly to their training by use of image encoders pretrained on ImageNet, supports claim A, and calls claim I into question.

Column legend (each group of 4 columns = before | after joint, pretraining | after joint, no pretraining | after separate):

- Columns i–iv: original / 40 classes, top 40
- Columns v–viii: original / bottom 960
- Columns ix–xii: original / original
- Columns xiii–xvi: 1000 classes / top 40
- Columns xvii–xx: 1000 classes / bottom 960
- Columns xxi–xxiv: 1000 classes / bottom 960

| data | net | model | # | i | ii | iii | iv | v | vi | vii | viii | ix | x | xi | xii | xiii | xiv | xv | xvi | xvii | xviii | xix | xx | xxi | xxii | xxiii | xxiv |
|---|---|---|---|---|---|---|---|---|---|---|---|---|---|---|---|---|---|---|---|---|---|---|---|---|---|---|---|
| their | EEG | Inception v3 | i | 2.6 | 3.0 | 2.5 | 55.5 | 2.2 | 2.6 | 2.5 | 55.5 | 2.5 | 3.1 | 2.5 | 2.3 | 0.1 | 0.1 | 0.1 | 55.5 | 0.1 | 0.5 | 0.0 | 55.5 | 0.1 | 0.1 | 0.1 | 2.3 |
| | | ResNet-101 | ii | 2.3 | 3.0 | 2.6 | 54.7 | 2.8 | 2.4 | 2.4 | 54.7 | 2.4 | 3.1 | 2.5 | 2.3 | 0.0 | 0.1 | 0.0 | 54.7 | 0.0 | 0.0 | 0.0 | 54.7 | 0.1 | 0.1 | 0.1 | 2.3 |
| | | DenseNet-161 | iii | 2.5 | 3.1 | 2.4 | 57.4 | 2.4 | 2.6 | 2.3 | 57.4 | 2.5 | 3.2 | 2.5 | 2.5 | 0.1 | 0.1 | 0.0 | 57.4 | 0.0 | 0.1 | 0.0 | 57.4 | 0.1 | 0.1 | 0.1 | 2.5 |
| | | AlexNet | iv | 2.3 | 2.5 | 2.4 | 57.3 | 2.4 | 2.6 | 2.6 | 57.3 | 2.4 | 2.5 | 2.6 | 2.5 | 0.1 | 0.1 | 0.1 | 57.3 | 0.0 | 0.1 | 0.1 | 57.3 | 0.1 | 0.1 | 0.1 | 2.5 |
| | image | Inception v3 | v | 99.7 | 43.2 | 2.7 | 3.4 | 99.4 | 13.5 | 2.9 | 2.2 | 45.5 | 29.4 | 2.7 | 2.5 | 92.8 | 16.5 | 0.1 | 0.0 | 98.9 | 4.0 | 0.0 | 0.0 | 12.8 | 9.3 | 0.0 | 0.0 |
| | | ResNet-101 | vi | 99.4 | 24.2 | 2.4 | 82.5 | 98.3 | 7.3 | 2.9 | 70.5 | 33.0 | 9.5 | 2.1 | 39.4 | 90.5 | 2.6 | 0.1 | 82.5 | 95.6 | 0.3 | 0.0 | 70.5 | 9.5 | 0.7 | 0.1 | 39.2 |
| | | DenseNet-161 | vii | 99.2 | 7.2 | 2.4 | 23.3 | 97.9 | 4.4 | 2.3 | 3.5 | 42.7 | 3.8 | 2.6 | 2.6 | 89.4 | 0.1 | 0.0 | 12.1 | 95.8 | 0.0 | 0.0 | 0.1 | 14.2 | 0.0 | 0.0 | 0.0 |
| | | AlexNet | viii | 95.8 | 2.6 | 2.7 | 18.7 | 85.8 | 2.6 | 2.7 | 19.1 | 34.1 | 2.6 | 2.4 | 2.4 | 78.4 | 0.0 | 0.3 | 18.7 | 72.5 | 0.0 | 0.3 | 19.1 | 12.5 | 0.0 | 0.0 | 2.4 |
| block | EEG | Inception v3 | ix | 2.4 | 6.8 | 1.9 | 77.0 | 1.8 | 7.1 | 2.0 | 69.8 | 2.6 | 3.6 | 2.5 | 32.1 | 0.1 | 0.4 | 0.0 | 77.0 | 0.0 | 0.2 | 0.1 | 69.8 | 0.0 | 0.2 | 0.1 | 28.0 |
| | | ResNet-101 | x | 1.6 | 3.5 | 2.2 | 76.6 | 1.1 | 4.2 | 2.5 | 70.5 | 2.4 | 2.8 | 2.2 | 28.1 | 0.0 | 0.1 | 0.1 | 76.6 | 0.0 | 0.1 | 0.1 | 70.5 | 0.0 | 0.0 | 0.1 | 24.0 |
| | | DenseNet-161 | xi | 1.4 | 4.1 | 2.0 | 77.0 | 1.2 | 3.8 | 1.9 | 72.2 | 2.5 | 3.5 | 2.7 | 25.9 | 0.1 | 0.1 | 0.0 | 77.0 | 0.0 | 0.0 | 0.0 | 72.2 | 0.1 | 0.1 | 0.1 | 22.3 |
| | | AlexNet | xii | 2.9 | 4.1 | 2.9 | 76.5 | 2.5 | 4.2 | 2.2 | 70.0 | 2.0 | 3.4 | 2.5 | 27.8 | 0.0 | 0.1 | 0.1 | 76.4 | 0.2 | 0.3 | 0.2 | 70.0 | 0.0 | 0.1 | 0.1 | 24.2 |
| | image | Inception v3 | xiii | 99.8 | 96.4 | 3.0 | 9.3 | 99.4 | 94.7 | 3.1 | 2.9 | 46.0 | 46.8 | 2.4 | 2.5 | 93.0 | 75.1 | 0.1 | 0.5 | 98.8 | 87.0 | 0.0 | 0.0 | 13.2 | 1.7 | 0.0 | 0.0 |
| | | ResNet-101 | xiv | 99.3 | 75.0 | 2.4 | 11.0 | 98.5 | 68.5 | 2.7 | 3.5 | 34.1 | 14.7 | 2.5 | 3.7 | 90.5 | 56.4 | 0.1 | 2.0 | 96.2 | 52.4 | 0.0 | 0.1 | 9.8 | 0.2 | 0.0 | 0.0 |
| | | DenseNet-161 | xv | 99.1 | 95.8 | 2.4 | 5.7 | 97.7 | 86.8 | 2.9 | 3.8 | 43.9 | 50.9 | 2.1 | 2.6 | 89.5 | 71.1 | 0.1 | 0.2 | 96.4 | 67.2 | 0.4 | 0.2 | 15.0 | 17.9 | 0.0 | 0.0 |
| | | AlexNet | xvi | 95.5 | 60.9 | 2.5 | 3.5 | 85.7 | 50.9 | 2.4 | 3.2 | 34.6 | 15.6 | 2.5 | 2.3 | 77.9 | 25.4 | 0.0 | 0.2 | 72.2 | 19.7 | 0.0 | 2.5 | 12.9 | 4.2 | 0.0 | 2.3 |
| randomized | EEG | Inception v3 | xvii | 2.6 | 2.5 | 2.5 | 1.6 | 2.6 | 2.9 | 2.6 | 2.5 | 3.0 | 2.2 | 2.7 | 1.6 | 0.1 | 0.1 | 0.1 | 1.5 | 0.1 | 0.1 | 0.0 | 2.5 | 0.2 | 0.1 | 0.2 | 1.4 |
| | | ResNet-101 | xviii | 2.6 | 2.4 | 2.8 | 2.1 | 2.8 | 2.7 | 2.9 | 2.3 | 2.6 | 2.7 | 3.0 | 1.8 | 0.0 | 0.1 | 0.1 | 1.9 | 0.0 | 0.1 | 0.2 | 2.3 | 0.1 | 0.2 | 0.1 | 1.6 |
| | | DenseNet-161 | xix | 2.1 | 2.2 | 2.3 | 1.5 | 2.5 | 2.3 | 2.5 | 2.0 | 3.1 | 2.0 | 1.9 | 1.9 | 0.1 | 0.1 | 0.1 | 1.5 | 0.0 | 0.0 | 0.1 | 2.0 | 0.1 | 0.1 | 0.1 | 1.7 |
| | | AlexNet | xx | 2.5 | 2.8 | 3.0 | 2.2 | 2.2 | 2.4 | 2.9 | 1.6 | 2.6 | 2.8 | 3.1 | 2.4 | 0.2 | 0.1 | 0.2 | 2.2 | 0.3 | 0.0 | 0.1 | 1.6 | 0.1 | 0.1 | 0.1 | 1.9 |
| | image | Inception v3 | xxi | 99.8 | 91.1 | 3.4 | 9.4 | 99.4 | 84.7 | 3.0 | 2.9 | 46.0 | 57.1 | 2.1 | 2.5 | 93.0 | 64.0 | 0.1 | 0.6 | 98.8 | 73.3 | 0.0 | 0.0 | 13.2 | 20.4 | 0.0 | 0.0 |
| | | ResNet-101 | xxii | 99.3 | 69.8 | 2.9 | 11.2 | 98.5 | 53.1 | 2.6 | 4.0 | 34.1 | 9.4 | 3.1 | 2.2 | 90.5 | 42.0 | 0.0 | 2.0 | 96.2 | 28.5 | 0.0 | 0.1 | 9.8 | 3.5 | 0.0 | 0.0 |
| | | DenseNet-161 | xxiii | 99.1 | 65.6 | 2.1 | 5.1 | 97.7 | 36.0 | 2.5 | 3.0 | 43.9 | 36.1 | 2.3 | 3.0 | 89.5 | 30.5 | 0.1 | 0.0 | 96.4 | 11.7 | 0.0 | 0.1 | 15.0 | 13.6 | 0.0 | 0.0 |
| | | AlexNet | xxiv | 95.5 | 3.6 | 2.4 | 3.5 | 85.7 | 3.6 | 2.5 | 3.5 | 34.6 | 2.5 | 2.5 | 2.5 | 77.9 | 0.0 | 0.0 | 2.6 | 72.2 | 0.0 | 0.0 | 2.6 | 12.9 | 0.0 | 0.0 | 2.5 |

Table 12: Image classification accuracy is very high with just the top 40 principal components both with 40 classes (≥ 85.7%; column v, for rows v–viii, xiii–xvi, and xxi–xxiv) and 1000 classes (≥72.2%; column xvii, for rows v–viii, xiii–xvi, and xxi–xxiv). This holds both for 'their' data, 'block' data, and 'randomized' data. This, together with Tables 10 and 11, demonstrates that they are giving class information implicitly to their training by use of image encoders pretrained on ImageNet, supports claim A, and calls claim I into question.

| | | | | 40 classes — original | | | | 40 classes — top 40 | | | | 40 classes — bottom 960 | | | | 1000 classes — original | | | | 1000 classes — top 40 | | | | 1000 classes — bottom 960 | | | |
|---|---|---|---|---|---|---|---|---|---|---|---|---|---|---|---|---|---|---|---|---|---|---|---|---|---|---|---|
| | | | | before | joint (pre) | joint (no-pre) | separate | before | joint (pre) | joint (no-pre) | separate | before | joint (pre) | joint (no-pre) | separate | before | joint (pre) | joint (no-pre) | separate | before | joint (pre) | joint (no-pre) | separate | before | joint (pre) | joint (no-pre) | separate |
| | | | | i | ii | iii | iv | v | vi | vii | viii | ix | x | xi | xii | xiii | xiv | xv | xvi | xvii | xviii | xix | xx | xxi | xxii | xxiii | xxiv |
| their | EEG | Inception v3 | i | 2.6 | 3.0 | 2.5 | 55.5 | 2.2 | 2.6 | 2.5 | 55.5 | 2.5 | 3.1 | 2.5 | 2.3 | 0.1 | 0.1 | 0.1 | 55.5 | 0.1 | 0.1 | 0.1 | 0.1 | 0.1 | 0.1 | 0.1 | 2.3 |
| | | ResNet-101 | ii | 2.3 | 3.0 | 2.6 | 54.7 | 2.8 | 2.4 | 2.4 | 54.7 | 2.4 | 3.1 | 2.4 | 2.3 | 0.0 | 0.1 | 0.1 | 54.7 | 0.0 | 0.0 | 0.0 | 0.0 | 0.1 | 0.1 | 0.1 | 2.3 |
| | | DenseNet-161 | iii | 2.5 | 3.1 | 2.4 | 57.4 | 2.4 | 2.6 | 2.3 | 57.4 | 2.5 | 3.2 | 2.4 | 2.5 | 0.1 | 0.1 | 0.1 | 57.4 | 0.0 | 0.1 | 0.1 | 0.2 | 0.1 | 0.1 | 0.1 | 2.5 |
| | | AlexNet | iv | 2.3 | 2.5 | 2.4 | 57.3 | 2.4 | 2.6 | 2.6 | 57.3 | 2.4 | 2.5 | 2.6 | 2.5 | 0.1 | 0.1 | 0.1 | 57.3 | 0.0 | 0.1 | 0.1 | 0.0 | 0.1 | 0.1 | 0.1 | 2.5 |
| | image | Inception v3 | v | 99.7 | 43.2 | 2.7 | 3.4 | 99.4 | 13.5 | 2.9 | 2.2 | 45.5 | 29.4 | 2.7 | 2.5 | 92.8 | 16.5 | 0.1 | 0.0 | 98.9 | 4.0 | 0.3 | 0.0 | 12.8 | 9.3 | 0.0 | 0.0 |
| | | ResNet-101 | vi | 99.4 | 24.2 | 2.4 | 82.5 | 98.3 | 7.3 | 2.9 | 70.5 | 33.0 | 9.5 | 2.1 | 39.4 | 90.5 | 2.6 | 0.0 | 82.5 | 95.6 | 0.3 | 0.0 | 70.5 | 9.5 | 0.7 | 0.0 | 39.2 |
| | | DenseNet-161 | vii | 99.2 | 7.2 | 2.4 | 23.3 | 97.9 | 4.4 | 2.3 | 3.5 | 42.7 | 3.8 | 2.6 | 2.6 | 89.4 | 0.1 | 0.0 | 12.1 | 95.8 | 0.0 | 0.0 | 0.1 | 14.2 | 0.0 | 0.0 | 0.0 |
| | | AlexNet | viii | 95.8 | 2.6 | 2.7 | 18.7 | 85.8 | 2.6 | 2.7 | 19.1 | 34.1 | 2.6 | 2.4 | 2.4 | 78.4 | 0.0 | 0.3 | 18.7 | 72.5 | 0.0 | 0.0 | 19.1 | 12.5 | 0.0 | 0.0 | 2.4 |
| block | EEG | Inception v3 | ix | 2.4 | 6.8 | 1.9 | 77.0 | 1.8 | 7.1 | 2.0 | 69.8 | 2.6 | 3.6 | 2.5 | 32.1 | 0.1 | 0.4 | 0.0 | 77.0 | 0.0 | 0.2 | 0.0 | 69.8 | 0.0 | 0.2 | 0.1 | 28.0 |
| | | ResNet-101 | x | 1.6 | 3.5 | 2.2 | 76.6 | 1.1 | 4.2 | 2.5 | 70.5 | 2.4 | 2.8 | 2.2 | 28.1 | 0.0 | 0.1 | 0.1 | 76.6 | 0.0 | 0.1 | 0.1 | 70.5 | 0.0 | 0.0 | 0.1 | 24.0 |
| | | DenseNet-161 | xi | 1.4 | 4.1 | 2.0 | 77.0 | 1.2 | 3.8 | 1.9 | 72.2 | 2.5 | 3.5 | 2.7 | 25.9 | 0.1 | 0.1 | 0.0 | 77.0 | 0.0 | 0.0 | 0.0 | 72.2 | 0.1 | 0.1 | 0.1 | 22.3 |
| | | AlexNet | xii | 2.9 | 4.1 | 2.9 | 76.5 | 2.5 | 4.2 | 2.2 | 70.0 | 2.0 | 3.4 | 2.5 | 27.8 | 0.0 | 0.1 | 0.1 | 76.4 | 0.0 | 0.3 | 0.2 | 70.0 | 0.1 | 0.0 | 0.1 | 24.2 |
| | image | Inception v3 | xiii | 99.8 | 96.4 | 3.0 | 9.3 | 99.4 | 94.7 | 3.1 | 2.9 | 46.0 | 46.8 | 2.4 | 2.5 | 93.0 | 75.1 | 0.1 | 0.5 | 98.8 | 87.0 | 0.0 | 0.0 | 13.2 | 1.7 | 0.0 | 0.0 |
| | | ResNet-101 | xiv | 99.3 | 75.0 | 2.4 | 11.0 | 98.5 | 68.5 | 2.7 | 3.5 | 34.1 | 14.7 | 2.5 | 3.7 | 90.5 | 56.4 | 0.0 | 2.0 | 96.2 | 52.4 | 0.4 | 0.1 | 9.8 | 0.2 | 0.0 | 0.0 |
| | | DenseNet-161 | xv | 99.1 | 95.8 | 2.5 | 5.7 | 97.7 | 86.8 | 2.9 | 3.8 | 43.9 | 50.9 | 2.1 | 2.6 | 89.5 | 71.1 | 0.1 | 0.2 | 96.4 | 67.2 | 0.0 | 0.2 | 15.0 | 17.9 | 0.0 | 0.0 |
| | | AlexNet | xvi | 95.5 | 60.9 | 2.5 | 3.5 | 85.7 | 50.9 | 2.4 | 3.2 | 34.6 | 15.6 | 2.5 | 2.3 | 77.9 | 25.4 | 0.0 | 2.5 | 72.2 | 19.7 | 0.0 | 2.5 | 12.9 | 4.2 | 0.2 | 2.3 |
| randomized | EEG | Inception v3 | xvii | 2.6 | 2.5 | 2.5 | 1.6 | 2.6 | 2.9 | 2.6 | 2.5 | 3.0 | 2.2 | 2.7 | 1.6 | 0.1 | 0.1 | 0.1 | 1.5 | 0.1 | 0.1 | 0.0 | 2.5 | 0.2 | 0.1 | 0.2 | 1.4 |
| | | ResNet-101 | xviii | 2.6 | 2.4 | 2.8 | 2.1 | 2.8 | 2.7 | 2.9 | 2.3 | 2.6 | 2.7 | 3.0 | 1.8 | 0.0 | 0.1 | 0.1 | 1.9 | 0.0 | 0.0 | 0.0 | 2.3 | 0.1 | 0.2 | 0.1 | 1.6 |
| | | DenseNet-161 | xix | 2.1 | 2.2 | 2.3 | 1.5 | 2.5 | 2.3 | 2.5 | 2.0 | 3.1 | 2.0 | 1.9 | 1.9 | 0.1 | 0.1 | 0.1 | 1.5 | 0.0 | 0.0 | 0.0 | 2.0 | 0.1 | 0.1 | 0.1 | 1.7 |
| | | AlexNet | xx | 2.5 | 2.8 | 3.0 | 2.2 | 2.2 | 2.4 | 2.9 | 1.6 | 2.6 | 2.8 | 3.1 | 2.4 | 0.2 | 0.1 | 0.2 | 2.2 | 0.3 | 0.0 | 0.0 | 1.6 | 0.1 | 0.1 | 0.1 | 1.9 |
| | image | Inception v3 | xxi | 99.8 | 91.1 | 3.4 | 9.4 | 99.4 | 84.7 | 3.0 | 2.9 | 46.0 | 57.1 | 2.1 | 2.5 | 93.0 | 64.0 | 0.1 | 0.6 | 98.8 | 73.3 | 0.0 | 0.0 | 13.2 | 20.4 | 0.0 | 0.0 |
| | | ResNet-101 | xxii | 99.3 | 69.8 | 2.9 | 11.2 | 98.5 | 53.1 | 2.6 | 4.0 | 34.1 | 9.4 | 3.1 | 2.2 | 90.5 | 42.0 | 0.0 | 2.0 | 96.2 | 28.5 | 0.0 | 0.1 | 9.8 | 3.5 | 0.0 | 0.0 |
| | | DenseNet-161 | xxiii | 99.1 | 65.6 | 2.1 | 5.1 | 97.7 | 36.0 | 2.5 | 3.0 | 43.9 | 36.1 | 2.3 | 3.0 | 89.5 | 30.5 | 0.1 | 0.0 | 96.4 | 11.7 | 0.0 | 0.0 | 15.0 | 13.6 | 0.0 | 0.0 |
| | | AlexNet | xxiv | 95.5 | 3.6 | 2.4 | 3.5 | 85.7 | 3.6 | 2.5 | 3.5 | 34.6 | 2.5 | 2.5 | 2.5 | 77.9 | 0.0 | 0.0 | 2.6 | 72.2 | 0.0 | 0.0 | 2.6 | 12.9 | 0.0 | 0.0 | 2.5 |

Table 13: Image classification accuracy is much lower than perfect (≤46.0%) before training when discarding the top 40 principal components (columns ix and xxi, for rows v–viii, xiii–xvi, and xxi–xxiv). This holds both for 40 classes and 1000 classes and both for 'their' data, 'block' data, and 'randomized' data. Thus there is much less class information outside the top 40 principal components. This demonstrates that they are giving little more than class information implicitly to their training by use of image encoders pretrained on ImageNet and further supports claim A and calls claim I into question.

| | | | 40 classes | | | | | | | | | | | | 1000 classes | | | | | | | | | | | |
| | | | original | | | | top 40 | | | | bottom 960 | | | | original | | | | top 40 | | | | bottom 960 | | | |
| | | | before | after joint | | after separate | before | after joint | | after separate | before | after joint | | after separate | before | after joint | | after separate | before | after joint | | after separate | before | after joint | | after separate |
| | | | | pretraining | no pretraining | | | pretraining | no pretraining | | | pretraining | no pretraining | | | pretraining | no pretraining | | | pretraining | no pretraining | | | pretraining | no pretraining | |
| | | | i | ii | iii | iv | v | vi | vii | viii | ix | x | xi | xii | xiii | xiv | xv | xvi | xvii | xviii | xix | xx | xxi | xxii | xxiii | xxiv |
|---|---|---|---|---|---|---|---|---|---|---|---|---|---|---|---|---|---|---|---|---|---|---|---|---|---|---|
| their | EEG | Inception v3 (i) | 2.6 | 3.0 | 2.5 | 55.5 | 2.2 | 2.6 | 2.5 | 55.5 | 2.5 | 3.1 | 2.5 | 2.3 | 0.1 | 0.1 | 0.1 | 55.5 | 0.1 | 0.1 | 0.1 | 55.5 | 0.1 | 0.1 | 0.1 | 2.3 |
| | | ResNet-101 (ii) | 2.3 | 3.0 | 2.6 | 54.7 | 2.8 | 2.4 | 2.4 | 54.7 | 2.4 | 3.1 | 2.4 | 2.3 | 0.0 | 0.1 | 0.1 | 54.7 | 0.0 | 0.0 | 0.0 | 54.7 | 0.1 | 0.1 | 0.1 | 2.3 |
| | | DenseNet-161 (iii) | 2.5 | 3.1 | 2.4 | 57.4 | 2.4 | 2.6 | 2.3 | 57.4 | 2.5 | 3.2 | 2.4 | 2.5 | 0.1 | 0.1 | 0.1 | 57.4 | 0.0 | 0.1 | 0.0 | 57.4 | 0.1 | 0.1 | 0.1 | 2.5 |
| | | AlexNet (iv) | 2.3 | 2.5 | 2.4 | 57.3 | 2.4 | 2.6 | 2.6 | 57.3 | 2.4 | 2.5 | 2.6 | 2.5 | 0.1 | 0.1 | 0.1 | 57.3 | 0.0 | 0.1 | 0.1 | 57.3 | 0.1 | 0.1 | 0.1 | 2.5 |
| | image | Inception v3 (v) | 99.7 | 43.2 | 2.7 | 3.4 | 99.4 | 13.5 | 2.9 | 2.2 | 45.5 | 29.4 | 2.7 | 2.5 | 92.8 | 16.5 | 0.0 | 0.0 | 98.9 | 4.0 | 0.0 | 0.0 | 12.8 | 9.3 | 0.0 | 0.0 |
| | | ResNet-101 (vi) | 99.4 | 24.2 | 2.4 | 82.5 | 98.3 | 7.3 | 2.9 | 70.5 | 33.0 | 9.5 | 2.1 | 39.4 | 90.5 | 2.6 | 0.0 | 82.5 | 95.6 | 0.3 | 0.3 | 70.5 | 9.5 | 0.7 | 0.0 | 39.2 |
| | | DenseNet-161 (vii) | 99.2 | 7.2 | 2.4 | 23.3 | 97.9 | 4.4 | 2.3 | 3.5 | 42.7 | 3.8 | 2.6 | 2.6 | 89.4 | 0.1 | 0.0 | 12.1 | 95.8 | 0.0 | 0.0 | 0.1 | 14.2 | 0.0 | 0.0 | 0.0 |
| | | AlexNet (viii) | 95.8 | 2.6 | 2.7 | 18.7 | 85.8 | 2.6 | 2.7 | 19.1 | 34.1 | 2.6 | 2.4 | 2.4 | 78.4 | 0.0 | 0.3 | 18.7 | 72.5 | 0.0 | 0.0 | 19.1 | 12.5 | 0.0 | 0.0 | 2.4 |
| block | EEG | Inception v3 (ix) | 2.4 | 6.8 | 1.9 | 77.0 | 1.8 | 7.1 | 2.0 | 69.8 | 2.6 | 3.6 | 2.5 | 32.1 | 0.1 | 0.4 | 0.0 | 77.0 | 0.0 | 0.2 | 0.1 | 69.8 | 0.0 | 0.2 | 0.1 | 28.0 |
| | | ResNet-101 (x) | 1.6 | 3.5 | 2.2 | 76.6 | 1.1 | 4.2 | 2.5 | 70.5 | 2.4 | 2.8 | 2.2 | 28.1 | 0.0 | 0.1 | 0.1 | 76.6 | 0.0 | 0.1 | 0.1 | 70.5 | 0.0 | 0.0 | 0.0 | 24.0 |
| | | DenseNet-161 (xi) | 1.4 | 4.1 | 2.0 | 77.0 | 1.2 | 3.8 | 1.9 | 72.2 | 2.5 | 3.5 | 2.7 | 25.9 | 0.1 | 0.1 | 0.0 | 77.0 | 0.0 | 0.0 | 0.0 | 72.2 | 0.1 | 0.1 | 0.0 | 22.3 |
| | | AlexNet (xii) | 2.9 | 4.1 | 2.9 | 76.5 | 2.5 | 4.2 | 2.2 | 70.0 | 2.0 | 3.4 | 2.5 | 27.8 | 0.0 | 0.1 | 0.1 | 76.4 | 0.0 | 0.3 | 0.2 | 70.0 | 0.0 | 0.0 | 0.1 | 24.2 |
| | image | Inception v3 (xiii) | 99.8 | 96.4 | 3.0 | 9.3 | 99.4 | 94.7 | 3.1 | 2.9 | 46.0 | 46.8 | 2.4 | 2.5 | 93.0 | 75.1 | 0.1 | 0.5 | 98.8 | 87.0 | 0.0 | 0.0 | 13.2 | 1.7 | 0.0 | 0.0 |
| | | ResNet-101 (xiv) | 99.3 | 75.0 | 2.4 | 11.0 | 98.5 | 68.5 | 2.7 | 3.5 | 34.1 | 14.7 | 2.5 | 3.7 | 90.5 | 56.4 | 0.0 | 2.0 | 96.4 | 52.4 | 0.0 | 0.1 | 9.8 | 0.2 | 0.0 | 0.0 |
| | | DenseNet-161 (xv) | 99.1 | 95.8 | 2.5 | 5.7 | 97.7 | 86.8 | 2.9 | 3.8 | 43.9 | 50.9 | 2.9 | 2.6 | 89.5 | 71.1 | 0.4 | 0.2 | 96.4 | 67.2 | 0.4 | 0.2 | 15.0 | 17.9 | 0.0 | 0.0 |
| | | AlexNet (xvi) | 95.5 | 60.9 | 2.5 | 3.5 | 85.7 | 50.9 | 2.4 | 3.2 | 34.6 | 15.6 | 2.5 | 2.3 | 77.9 | 25.4 | 0.0 | 2.5 | 72.2 | 19.7 | 0.0 | 2.5 | 12.9 | 4.2 | 0.2 | 2.3 |
| randomized | EEG | Inception v3 (xvii) | 2.6 | 2.5 | 2.5 | 1.6 | 2.6 | 2.9 | 2.6 | 2.5 | 3.0 | 2.2 | 2.7 | 1.6 | 0.1 | 0.1 | 0.1 | 1.5 | 0.1 | 0.1 | 0.2 | 2.5 | 0.2 | 0.1 | 0.1 | 1.4 |
| | | ResNet-101 (xviii) | 2.6 | 2.4 | 2.8 | 2.1 | 2.8 | 2.7 | 2.9 | 2.3 | 2.6 | 2.7 | 3.0 | 1.8 | 0.0 | 0.1 | 0.1 | 1.9 | 0.0 | 0.0 | 0.2 | 2.3 | 0.1 | 0.2 | 0.1 | 1.6 |
| | | DenseNet-161 (xix) | 2.1 | 2.2 | 2.3 | 1.5 | 2.5 | 2.3 | 2.5 | 2.0 | 3.1 | 2.0 | 1.9 | 1.9 | 0.1 | 0.1 | 0.1 | 1.5 | 0.0 | 0.0 | 0.1 | 2.0 | 0.1 | 0.1 | 0.1 | 1.7 |
| | | AlexNet (xx) | 2.5 | 2.8 | 3.0 | 2.2 | 2.2 | 2.4 | 2.9 | 1.6 | 2.6 | 2.8 | 3.1 | 2.4 | 0.2 | 0.1 | 0.2 | 2.2 | 0.3 | 0.0 | 0.1 | 1.6 | 0.1 | 0.1 | 0.1 | 1.9 |
| | image | Inception v3 (xxi) | 99.8 | 91.1 | 3.4 | 9.4 | 99.4 | 84.7 | 3.0 | 2.9 | 46.0 | 57.1 | 2.1 | 2.5 | 93.0 | 64.0 | 0.1 | 0.6 | 98.8 | 73.3 | 0.0 | 0.0 | 13.2 | 20.4 | 0.0 | 0.0 |
| | | ResNet-101 (xxii) | 99.3 | 69.8 | 2.9 | 11.2 | 98.5 | 53.1 | 2.6 | 4.0 | 34.1 | 9.4 | 3.1 | 2.2 | 90.5 | 42.0 | 0.0 | 2.0 | 96.2 | 28.5 | 0.0 | 0.1 | 9.8 | 3.5 | 0.0 | 0.0 |
| | | DenseNet-161 (xxiii) | 99.1 | 65.6 | 2.1 | 5.1 | 97.7 | 36.0 | 2.5 | 3.0 | 43.9 | 36.1 | 2.3 | 3.0 | 89.5 | 30.5 | 0.1 | 0.0 | 96.4 | 11.7 | 0.0 | 0.0 | 15.0 | 13.6 | 0.0 | 0.0 |
| | | AlexNet (xxiv) | 95.5 | 3.6 | 2.4 | 3.5 | 85.7 | 3.6 | 2.5 | 3.5 | 34.6 | 2.5 | 2.5 | 2.5 | 77.9 | 0.0 | 0.0 | 2.6 | 72.2 | 0.0 | 0.0 | 2.6 | 12.9 | 0.0 | 0.0 | 2.5 |

Table 14: With only five exceptions, all in the bottom 960 principal components, image classification accuracy decreases after joint training with pretraining (compare columns ii, vi, x, xiv, xviii, and xxii to i, v, ix,xiii, xvii, and xxi, respectively, for rows v–viii, xiii–xvi, and xxi–xxiv). This holds both for 40 classes and 1000 classes, both for 'their' data, 'block' data, and 'randomized' data, and whether using all components or just the 40 principal components. This suggests that joint training is hurting, not helping, and supports claim C(2).

Columns i–xii belong to "40 classes"; columns xiii–xxiv belong to "1000 classes". Within each, the four-column blocks are: original (i–iv, xiii–xvi), top 40 (v–viii, xvii–xx), bottom 960 (ix–xii, xxi–xxiv). Within each block the sub-columns are: before / after joint (pretraining) / after joint (no pretraining) / after separate.

| | | | Row | i | ii | iii | iv | v | vi | vii | viii | ix | x | xi | xii | xiii | xiv | xv | xvi | xvii | xviii | xix | xx | xxi | xxii | xxiii | xxiv |
|---|---|---|---|---|---|---|---|---|---|---|---|---|---|---|---|---|---|---|---|---|---|---|---|---|---|---|---|
| their | EEG | Inception v3 | 1 | 2.6 | 3.0 | 2.5 | 55.5 | 2.2 | 2.6 | 2.5 | 55.5 | 2.5 | 3.1 | 2.5 | 2.3 | 0.1 | 0.1 | 0.1 | 55.5 | 0.1 | 0.1 | 0.1 | 55.5 | 0.1 | 0.1 | 0.1 | 2.3 |
| | | ResNet-101 | ii | 2.3 | 3.0 | 2.6 | 54.7 | 2.8 | 2.4 | 2.4 | 54.7 | 2.4 | 3.1 | 2.5 | 2.3 | 0.0 | 0.1 | 0.1 | 54.7 | 0.0 | 0.0 | 0.1 | 54.7 | 0.1 | 0.1 | 0.1 | 2.3 |
| | | DenseNet-161 | iii | 2.5 | 3.1 | 2.4 | 57.4 | 2.4 | 2.6 | 2.3 | 57.4 | 2.5 | 3.2 | 2.4 | 2.5 | 0.1 | 0.1 | 0.1 | 57.4 | 0.0 | 0.1 | 0.0 | 57.4 | 0.1 | 0.1 | 0.1 | 2.5 |
| | | AlexNet | iv | 2.3 | 2.5 | 2.4 | 57.3 | 2.4 | 2.6 | 2.6 | 57.3 | 2.4 | 2.5 | 2.6 | 2.5 | 0.1 | 0.1 | 0.1 | 57.3 | 0.0 | 0.1 | 0.1 | 57.3 | 0.1 | 0.1 | 0.1 | 2.5 |
| | image | Inception v3 | v | 99.7 | 43.2 | 2.7 | 3.4 | 99.4 | 13.5 | 2.9 | 2.2 | 45.5 | 29.4 | 2.7 | 2.5 | 92.8 | 16.5 | 0.1 | 0.0 | 98.9 | 4.0 | 0.0 | 0.0 | 12.8 | 9.3 | 0.0 | 0.0 |
| | | ResNet-101 | vi | 99.4 | 24.2 | 2.4 | 82.5 | 98.3 | 7.3 | 2.9 | 70.5 | 33.0 | 9.5 | 2.1 | 39.4 | 90.5 | 2.6 | 0.0 | 82.5 | 95.6 | 0.3 | 0.0 | 70.5 | 9.5 | 0.7 | 0.0 | 39.2 |
| | | DenseNet-161 | vii | 99.2 | 7.2 | 2.4 | 23.3 | 97.9 | 4.4 | 2.3 | 3.5 | 42.7 | 3.8 | 2.6 | 2.6 | 89.4 | 0.1 | 0.0 | 12.1 | 95.8 | 0.0 | 0.0 | 0.1 | 14.2 | 0.0 | 0.0 | 0.0 |
| | | AlexNet | viii | 95.8 | 2.6 | 2.7 | 18.7 | 85.8 | 2.6 | 2.7 | 19.1 | 34.1 | 2.6 | 2.4 | 2.4 | 78.4 | 0.0 | 0.3 | 18.7 | 72.5 | 0.0 | 0.3 | 19.1 | 12.5 | 0.0 | 0.0 | 2.4 |
| block | EEG | Inception v3 | ix | 2.4 | 6.8 | 1.9 | 77.0 | 1.8 | 7.1 | 2.0 | 69.8 | 2.6 | 3.6 | 2.5 | 32.1 | 0.1 | 0.4 | 0.0 | 77.0 | 0.0 | 0.2 | 0.1 | 69.8 | 0.0 | 0.2 | 0.1 | 28.0 |
| | | ResNet-101 | x | 1.6 | 3.5 | 2.2 | 76.6 | 1.1 | 4.2 | 2.5 | 70.5 | 2.4 | 2.8 | 2.2 | 28.1 | 0.0 | 0.1 | 0.1 | 76.6 | 0.0 | 0.1 | 0.1 | 70.5 | 0.0 | 0.0 | 0.0 | 24.0 |
| | | DenseNet-161 | xi | 1.4 | 4.1 | 2.0 | 77.0 | 1.2 | 3.8 | 1.9 | 72.2 | 2.5 | 3.5 | 2.7 | 25.9 | 0.1 | 0.1 | 0.0 | 77.0 | 0.0 | 0.0 | 0.0 | 72.2 | 0.1 | 0.1 | 0.1 | 22.3 |
| | | AlexNet | xii | 2.9 | 4.1 | 2.9 | 76.5 | 2.5 | 4.2 | 2.2 | 70.0 | 2.0 | 3.4 | 2.5 | 27.8 | 0.0 | 0.1 | 0.1 | 76.4 | 0.0 | 0.3 | 0.2 | 70.0 | 0.0 | 0.1 | 0.1 | 24.2 |
| | image | Inception v3 | xiii | 99.8 | 96.4 | 3.0 | 9.3 | 99.4 | 94.7 | 3.1 | 2.9 | 46.0 | 46.8 | 2.4 | 2.5 | 93.0 | 75.1 | 0.1 | 0.5 | 98.8 | 87.0 | 0.0 | 0.0 | 13.2 | 11.7 | 0.0 | 0.0 |
| | | ResNet-101 | xiv | 99.3 | 75.0 | 2.4 | 11.0 | 98.5 | 68.5 | 2.7 | 3.5 | 34.1 | 14.7 | 2.5 | 3.7 | 90.5 | 56.4 | 0.0 | 2.0 | 96.2 | 52.4 | 0.1 | 0.1 | 9.8 | 0.2 | 0.0 | 0.0 |
| | | DenseNet-161 | xv | 99.1 | 95.8 | 2.5 | 5.7 | 97.7 | 86.8 | 2.9 | 3.8 | 43.9 | 50.9 | 2.1 | 2.6 | 89.5 | 71.1 | 0.1 | 0.2 | 96.4 | 67.2 | 0.4 | 0.2 | 15.0 | 17.9 | 0.0 | 0.0 |
| | | AlexNet | xvi | 95.5 | 60.9 | 2.5 | 3.5 | 85.7 | 50.9 | 2.4 | 3.2 | 34.6 | 15.6 | 2.5 | 2.3 | 77.9 | 25.4 | 0.0 | 2.6 | 72.2 | 19.7 | 0.0 | 2.5 | 12.9 | 4.2 | 0.2 | 2.3 |
| randomized | EEG | Inception v3 | xvii | 2.6 | 2.5 | 2.5 | 1.6 | 2.6 | 2.9 | 2.6 | 2.5 | 3.0 | 2.2 | 2.7 | 1.6 | 0.1 | 0.1 | 0.1 | 1.5 | 0.1 | 0.1 | 0.0 | 2.5 | 0.2 | 0.1 | 0.2 | 1.4 |
| | | ResNet-101 | xviii | 2.6 | 2.4 | 2.8 | 2.1 | 2.8 | 2.7 | 2.9 | 2.3 | 2.6 | 2.7 | 3.0 | 1.8 | 0.0 | 0.1 | 0.1 | 1.9 | 0.0 | 0.1 | 0.2 | 2.3 | 0.1 | 0.2 | 0.1 | 1.6 |
| | | DenseNet-161 | xix | 2.1 | 2.2 | 2.3 | 1.5 | 2.5 | 2.3 | 2.5 | 2.0 | 3.1 | 2.0 | 1.9 | 1.9 | 0.1 | 0.1 | 0.1 | 1.5 | 0.0 | 0.0 | 0.1 | 2.0 | 0.1 | 0.1 | 0.1 | 1.7 |
| | | AlexNet | xx | 2.5 | 2.8 | 3.0 | 2.2 | 2.2 | 2.4 | 2.9 | 1.6 | 2.6 | 2.8 | 3.1 | 2.4 | 0.2 | 0.1 | 0.2 | 2.2 | 0.3 | 0.0 | 0.0 | 1.6 | 0.1 | 0.1 | 0.1 | 1.9 |
| | image | Inception v3 | xxi | 99.8 | 91.1 | 3.4 | 9.4 | 99.4 | 84.7 | 3.0 | 2.9 | 46.0 | 57.1 | 2.1 | 2.5 | 93.0 | 64.0 | 0.1 | 0.6 | 98.8 | 73.3 | 0.0 | 0.0 | 13.2 | 20.4 | 0.0 | 0.0 |
| | | ResNet-101 | xxii | 99.3 | 69.8 | 2.9 | 11.2 | 98.5 | 53.1 | 2.6 | 4.0 | 34.1 | 9.4 | 3.1 | 2.2 | 90.5 | 42.0 | 0.0 | 2.0 | 96.2 | 28.5 | 0.2 | 0.1 | 9.8 | 3.5 | 0.0 | 0.0 |
| | | DenseNet-161 | xxiii | 99.1 | 65.6 | 2.1 | 5.1 | 97.7 | 36.0 | 2.5 | 3.0 | 43.9 | 36.1 | 2.3 | 3.0 | 89.5 | 30.5 | 0.1 | 0.0 | 96.4 | 11.7 | 0.0 | 0.0 | 15.0 | 13.6 | 0.0 | 0.0 |
| | | AlexNet | xxiv | 95.5 | 3.6 | 2.4 | 3.5 | 85.7 | 3.6 | 2.5 | 3.5 | 34.6 | 2.5 | 2.5 | 2.5 | 77.9 | 0.0 | 0.0 | 2.6 | 72.2 | 0.0 | 0.0 | 2.6 | 12.9 | 0.0 | 0.0 | 2.5 |

Table 15: With the exception of a small number of cases that are marginally above chance, EEG classification accuracy is at chance except after separate training on confounded data either on all components or the top 40 components (columns iv, viii, xvi, and xx, for rows i–iv and ix–xii). This, together with Table 16, suggests that separate training is able to extract class from confounded data but not nonconfounded data and that joint training is not able to extract class from any data, and supports claims C(1) and D.

Column groups (each group: before | after joint pretraining | after joint no pretraining | after separate):
- 40 classes, original: i–iv
- 40 classes, top 40: v–viii
- bottom 960: ix–xii
- 1000 classes, original: xiii–xvi
- 1000 classes, top 40: xvii–xx
- bottom 960: xxi–xxiv

| | | Model | # | i | ii | iii | iv | v | vi | vii | viii | ix | x | xi | xii | xiii | xiv | xv | xvi | xvii | xviii | xix | xx | xxi | xxii | xxiii | xxiv |
|---|---|---|---|---|---|---|---|---|---|---|---|---|---|---|---|---|---|---|---|---|---|---|---|---|---|---|---|
| their | EEG | Inception v3 | i | 2.6 | 3.0 | 2.5 | 55.5 | 2.2 | 2.6 | 2.5 | 55.5 | 2.5 | 3.1 | 2.5 | 2.3 | 0.1 | 0.1 | 0.1 | 55.5 | 0.1 | 0.1 | 0.1 | 55.5 | 0.1 | 0.1 | 0.1 | 2.3 |
| | | ResNet-101 | ii | 2.3 | 3.0 | 2.6 | 54.7 | 2.8 | 2.4 | 2.4 | 54.7 | 2.4 | 3.1 | 2.5 | 2.3 | 0.0 | 0.1 | 0.1 | 54.7 | 0.0 | 0.0 | 0.1 | 54.7 | 0.1 | 0.1 | 0.1 | 2.3 |
| | | DenseNet-161 | iii | 2.5 | 3.1 | 2.4 | 57.4 | 2.4 | 2.6 | 2.3 | 57.4 | 2.5 | 3.2 | 2.4 | 2.5 | 0.1 | 0.1 | 0.1 | 57.4 | 0.1 | 0.1 | 0.0 | 57.4 | 0.1 | 0.1 | 0.1 | 2.5 |
| | | AlexNet | iv | 2.3 | 2.5 | 2.4 | 57.3 | 2.4 | 2.6 | 2.6 | 57.3 | 2.4 | 2.5 | 2.6 | 2.5 | 0.1 | 0.1 | 0.1 | 57.3 | 0.0 | 0.1 | 0.1 | 57.3 | 0.1 | 0.1 | 0.1 | 2.5 |
| | image | Inception v3 | v | 99.7 | 43.2 | 2.7 | 3.4 | 99.4 | 13.5 | 2.9 | 2.2 | 45.5 | 29.4 | 2.7 | 2.5 | 92.8 | 16.5 | 0.1 | 0.0 | 98.9 | 4.0 | 0.0 | 0.0 | 12.8 | 9.3 | 0.0 | 0.0 |
| | | ResNet-101 | vi | 99.4 | 24.2 | 2.4 | 82.5 | 98.3 | 7.3 | 2.9 | 70.5 | 33.0 | 9.5 | 2.1 | 39.4 | 90.5 | 2.6 | 0.0 | 82.5 | 95.6 | 0.3 | 0.0 | 70.5 | 9.5 | 0.7 | 0.0 | 39.2 |
| | | DenseNet-161 | vii | 99.2 | 7.2 | 2.4 | 23.3 | 97.9 | 4.4 | 2.3 | 3.5 | 42.7 | 3.8 | 2.6 | 2.6 | 89.4 | 0.1 | 0.1 | 12.1 | 95.8 | 0.0 | 0.0 | 0.1 | 14.2 | 0.0 | 0.0 | 0.0 |
| | | AlexNet | viii | 95.8 | 2.6 | 2.7 | 18.7 | 85.8 | 2.6 | 2.7 | 19.1 | 34.1 | 2.6 | 2.4 | 2.4 | 78.4 | 0.0 | 0.3 | 18.7 | 72.5 | 0.0 | 0.3 | 19.1 | 12.5 | 0.0 | 0.0 | 2.4 |
| block | EEG | Inception v3 | ix | 2.4 | 6.8 | 1.9 | 77.0 | 1.8 | 7.1 | 2.0 | 69.8 | 2.6 | 3.6 | 2.5 | 32.1 | 0.1 | 0.4 | 0.0 | 77.0 | 0.0 | 0.2 | 0.1 | 69.8 | 0.0 | 0.2 | 0.1 | 28.0 |
| | | ResNet-101 | x | 1.6 | 3.5 | 2.2 | 76.6 | 1.1 | 4.2 | 2.5 | 70.5 | 2.4 | 2.8 | 2.2 | 28.1 | 0.0 | 0.1 | 0.1 | 76.6 | 0.0 | 0.1 | 0.1 | 70.5 | 0.0 | 0.0 | 0.1 | 24.0 |
| | | DenseNet-161 | xi | 1.4 | 4.1 | 2.0 | 77.0 | 1.2 | 3.8 | 1.9 | 72.2 | 2.5 | 3.5 | 2.7 | 25.9 | 0.1 | 0.1 | 0.0 | 77.0 | 0.0 | 0.0 | 0.0 | 72.2 | 0.1 | 0.1 | 0.1 | 22.3 |
| | | AlexNet | xii | 2.9 | 4.1 | 2.9 | 76.5 | 2.5 | 4.2 | 2.2 | 70.0 | 2.0 | 3.4 | 2.5 | 27.8 | 0.0 | 0.1 | 0.1 | 76.4 | 0.0 | 0.3 | 0.2 | 70.0 | 0.1 | 0.0 | 0.1 | 24.2 |
| | image | Inception v3 | xiii | 99.8 | 96.4 | 3.0 | 9.3 | 99.4 | 94.7 | 3.1 | 2.9 | 46.0 | 46.8 | 2.4 | 2.5 | 93.0 | 75.1 | 0.0 | 0.5 | 98.8 | 87.0 | 0.0 | 0.0 | 13.2 | 1.7 | 0.0 | 0.0 |
| | | ResNet-101 | xiv | 99.3 | 75.0 | 2.4 | 11.0 | 98.5 | 68.5 | 2.7 | 3.5 | 34.1 | 14.7 | 2.5 | 3.7 | 90.5 | 56.4 | 0.0 | 2.0 | 96.2 | 52.4 | 0.2 | 0.1 | 9.8 | 0.2 | 0.0 | 0.0 |
| | | DenseNet-161 | xv | 99.1 | 95.8 | 2.5 | 5.7 | 97.7 | 86.8 | 2.9 | 3.8 | 43.9 | 50.9 | 2.1 | 2.6 | 89.5 | 71.1 | 0.1 | 0.2 | 96.4 | 67.2 | 0.4 | 0.2 | 15.0 | 17.9 | 0.0 | 0.0 |
| | | AlexNet | xvi | 95.5 | 60.9 | 2.5 | 3.5 | 85.7 | 50.9 | 2.4 | 3.2 | 34.6 | 15.6 | 2.5 | 2.3 | 77.9 | 25.4 | 0.0 | 0.0 | 72.2 | 19.7 | 0.0 | 2.5 | 12.9 | 4.2 | 0.0 | 2.3 |
| randomized | EEG | Inception v3 | xvii | 2.6 | 2.5 | 2.5 | 1.6 | 2.6 | 2.9 | 2.6 | 2.5 | 3.0 | 2.2 | 2.7 | 1.6 | 0.1 | 0.1 | 0.1 | 1.5 | 0.1 | 0.1 | 0.0 | 2.5 | 0.2 | 0.1 | 0.2 | 1.4 |
| | | ResNet-101 | xviii | 2.6 | 2.4 | 2.8 | 2.1 | 2.8 | 2.7 | 2.9 | 2.3 | 2.6 | 2.7 | 3.0 | 1.8 | 0.0 | 0.1 | 0.1 | 1.9 | 0.0 | 0.1 | 0.2 | 2.3 | 0.2 | 0.2 | 0.1 | 1.6 |
| | | DenseNet-161 | xix | 2.1 | 2.2 | 2.3 | 1.5 | 2.5 | 2.3 | 2.5 | 2.0 | 3.1 | 2.0 | 1.9 | 1.9 | 0.1 | 0.1 | 0.1 | 1.5 | 0.0 | 0.0 | 0.1 | 2.0 | 0.1 | 0.1 | 0.1 | 1.7 |
| | | AlexNet | xx | 2.5 | 2.8 | 3.0 | 2.2 | 2.2 | 2.4 | 2.9 | 1.6 | 2.6 | 2.8 | 3.1 | 2.4 | 0.2 | 0.1 | 0.2 | 2.2 | 0.3 | 0.0 | 0.1 | 1.6 | 0.1 | 0.1 | 0.1 | 1.9 |
| | image | Inception v3 | xxi | 99.8 | 91.1 | 3.4 | 9.4 | 99.4 | 84.7 | 3.0 | 2.9 | 46.0 | 57.1 | 2.1 | 2.5 | 93.0 | 64.0 | 0.1 | 0.6 | 98.8 | 73.3 | 0.0 | 0.0 | 13.2 | 20.4 | 0.0 | 0.0 |
| | | ResNet-101 | xxii | 99.3 | 69.8 | 2.9 | 11.2 | 98.5 | 53.1 | 2.6 | 4.0 | 34.1 | 9.4 | 3.1 | 2.2 | 90.5 | 42.0 | 0.0 | 2.0 | 96.2 | 28.5 | 0.2 | 0.1 | 9.8 | 3.5 | 0.0 | 0.1 |
| | | DenseNet-161 | xxiii | 99.1 | 65.6 | 2.1 | 5.1 | 97.7 | 36.0 | 2.5 | 3.0 | 43.9 | 36.1 | 2.3 | 3.0 | 89.5 | 30.5 | 0.1 | 0.0 | 96.4 | 11.7 | 0.0 | 0.0 | 15.0 | 13.6 | 0.0 | 0.0 |
| | | AlexNet | xxiv | 95.5 | 3.6 | 2.4 | 3.5 | 85.7 | 3.6 | 2.5 | 3.5 | 34.6 | 2.5 | 2.5 | 2.5 | 77.9 | 0.0 | 0.0 | 2.6 | 72.2 | 0.0 | 0.0 | 2.6 | 12.9 | 0.0 | 0.0 | 2.5 |

Table 16: EEG classification accuracy is also above chance after separate training on 'block' data in the bottom 960 components (columns xii and xxiv, for rows ix—xii). This, together with Table 15, suggests that separate training is able to extract class from confounded data but not nonconfounded data and that joint training is not able to extract class from any data, and supports claims C(1) and D.

| | | | | original | | | | 40 classes top-40 | | | | bottom 960 | | | | original | | | | 1000 classes top-40 | | | | bottom 960 | | | |
| | | | | before | after joint pretraining | after joint no pretraining | after separate | before | after joint pretraining | after joint no pretraining | after separate | before | after joint pretraining | after joint no pretraining | after separate | before | after joint pretraining | after joint no pretraining | after separate | before | after joint pretraining | after joint no pretraining | after separate | before | after joint pretraining | after joint no pretraining | after separate |
|---|---|---|---|---|---|---|---|---|---|---|---|---|---|---|---|---|---|---|---|---|---|---|---|---|---|---|
| | | | | i | ii | iii | iv | v | vi | vii | viii | ix | x | xi | xii | xiii | xiv | xv | xvi | xvii | xviii | xix | xx | xxi | xxii | xxiii | xxiv |
| their | EEG | Inception-v3 | i | 2.6 | 3.0 | 2.5 | 55.5 | 2.2 | 2.6 | 2.5 | 55.5 | 2.5 | 3.1 | 2.5 | 2.3 | 0.1 | 0.1 | 0.1 | 55.5 | 0.1 | 0.1 | 0.1 | 55.5 | 0.1 | 0.1 | 0.1 | 2.3 |
| | | ResNet-101 | ii | 2.3 | 3.0 | 2.6 | 54.7 | 2.8 | 2.4 | 2.4 | 54.7 | 2.4 | 3.1 | 2.5 | 2.3 | 0.0 | 0.1 | 0.1 | 54.7 | 0.0 | 0.0 | 0.0 | 54.7 | 0.1 | 0.1 | 0.1 | 2.3 |
| | | DenseNet-161 | iii | 2.5 | 3.1 | 2.4 | 57.4 | 2.4 | 2.6 | 2.3 | 57.4 | 2.5 | 3.2 | 2.4 | 2.5 | 0.1 | 0.1 | 0.1 | 57.4 | 0.0 | 0.1 | 0.0 | 57.4 | 0.1 | 0.1 | 0.1 | 2.5 |
| | | AlexNet | iv | 2.3 | 2.5 | 2.4 | 57.3 | 2.4 | 2.6 | 2.6 | 57.3 | 2.4 | 2.5 | 2.6 | 2.5 | 0.1 | 0.1 | 0.1 | 57.3 | 0.0 | 0.1 | 0.1 | 57.3 | 0.1 | 0.1 | 0.1 | 2.5 |
| | image | Inception-v3 | v | 99.7 | 43.2 | 2.7 | 3.4 | 99.4 | 13.5 | 2.9 | 2.2 | 45.5 | 29.4 | 2.7 | 2.5 | 92.8 | 16.5 | 0.0 | 0.0 | 98.9 | 4.0 | 0.0 | 0.0 | 12.8 | 9.3 | 0.0 | 0.0 |
| | | ResNet-101 | vi | 99.4 | 24.2 | 2.4 | 82.5 | 98.3 | 7.3 | 2.9 | 70.5 | 33.0 | 9.5 | 2.1 | 39.4 | 90.5 | 2.6 | 0.0 | 82.5 | 95.6 | 0.3 | 0.0 | 70.5 | 9.5 | 0.7 | 0.0 | 39.2 |
| | | DenseNet-161 | vii | 99.2 | 7.2 | 2.4 | 23.3 | 97.9 | 4.4 | 2.3 | 3.5 | 42.7 | 3.8 | 2.6 | 2.6 | 89.4 | 0.1 | 0.0 | 12.1 | 95.8 | 0.0 | 0.0 | 0.1 | 14.2 | 0.1 | 0.0 | 0.0 |
| | | AlexNet | viii | 95.8 | 2.6 | 2.7 | 18.7 | 85.8 | 2.6 | 2.7 | 19.1 | 34.1 | 2.6 | 2.4 | 2.4 | 78.4 | 0.0 | 0.3 | 18.7 | 72.5 | 0.0 | 0.3 | 19.1 | 12.5 | 0.0 | 0.0 | 2.4 |
| block | EEG | Inception-v3 | ix | 2.4 | 6.8 | 1.9 | 77.0 | 1.8 | 7.1 | 2.0 | 69.8 | 2.6 | 3.6 | 2.5 | 32.1 | 0.1 | 0.4 | 0.0 | 77.0 | 0.0 | 0.2 | 0.1 | 69.8 | 0.0 | 0.2 | 0.1 | 28.0 |
| | | ResNet-101 | x | 1.6 | 3.5 | 2.2 | 76.6 | 1.1 | 4.2 | 2.5 | 70.5 | 2.4 | 2.8 | 2.2 | 28.1 | 0.0 | 0.1 | 0.1 | 76.6 | 0.0 | 0.1 | 0.1 | 70.5 | 0.0 | 0.1 | 0.1 | 24.0 |
| | | DenseNet-161 | xi | 1.4 | 4.1 | 2.0 | 77.0 | 1.2 | 3.8 | 1.9 | 72.2 | 2.5 | 3.5 | 2.7 | 25.9 | 0.1 | 0.1 | 0.0 | 77.0 | 0.0 | 0.0 | 0.0 | 72.2 | 0.1 | 0.1 | 0.0 | 22.3 |
| | | AlexNet | xii | 2.9 | 4.1 | 2.9 | 76.5 | 2.5 | 4.2 | 2.2 | 70.0 | 2.0 | 3.4 | 2.5 | 27.8 | 0.0 | 0.1 | 0.1 | 76.4 | 0.0 | 0.3 | 0.2 | 70.0 | 0.0 | 0.1 | 0.1 | 24.2 |
| | image | Inception-v3 | xiii | 99.8 | 96.4 | 3.0 | 9.3 | 99.4 | 94.7 | 3.1 | 2.9 | 46.0 | 46.8 | 2.4 | 2.5 | 93.0 | 75.1 | 0.1 | 0.5 | 98.8 | 87.0 | 0.0 | 0.0 | 13.2 | 1.7 | 0.0 | 0.0 |
| | | ResNet-101 | xiv | 99.3 | 75.0 | 2.4 | 11.0 | 98.5 | 68.5 | 2.7 | 3.5 | 34.1 | 14.7 | 2.5 | 3.7 | 90.5 | 56.4 | 0.0 | 2.0 | 96.2 | 52.4 | 0.0 | 0.1 | 9.8 | 0.2 | 0.0 | 0.0 |
| | | DenseNet-161 | xv | 99.1 | 95.8 | 2.5 | 5.7 | 97.7 | 86.8 | 2.9 | 3.8 | 43.9 | 50.9 | 2.1 | 2.6 | 89.5 | 71.1 | 0.4 | 0.2 | 96.4 | 67.2 | 0.4 | 0.2 | 15.0 | 17.9 | 0.0 | 0.0 |
| | | AlexNet | xvi | 95.5 | 60.9 | 2.5 | 3.5 | 85.7 | 50.9 | 2.4 | 3.2 | 34.6 | 15.6 | 2.5 | 2.3 | 77.9 | 25.4 | 0.0 | 0.2 | 72.2 | 19.7 | 0.0 | 2.5 | 12.9 | 4.2 | 0.0 | 2.3 |
| randomized | EEG | Inception-v3 | xvii | 2.6 | 2.5 | 2.5 | 1.6 | 2.6 | 2.9 | 2.6 | 2.5 | 3.0 | 2.2 | 2.7 | 1.6 | 0.1 | 0.1 | 0.0 | 1.5 | 0.1 | 0.1 | 0.0 | 2.5 | 0.2 | 0.1 | 0.2 | 1.4 |
| | | ResNet-101 | xviii | 2.6 | 2.4 | 2.8 | 2.1 | 2.8 | 2.7 | 2.9 | 2.3 | 2.6 | 2.7 | 3.0 | 1.8 | 0.0 | 0.1 | 0.2 | 1.9 | 0.0 | 0.2 | 0.2 | 2.3 | 0.1 | 0.2 | 0.1 | 1.6 |
| | | DenseNet-161 | xix | 2.1 | 2.2 | 2.3 | 1.5 | 2.5 | 2.3 | 2.5 | 2.0 | 3.1 | 2.0 | 1.9 | 1.9 | 0.1 | 0.1 | 0.1 | 1.5 | 0.0 | 0.0 | 0.1 | 2.0 | 0.1 | 0.1 | 0.1 | 1.7 |
| | | AlexNet | xx | 2.5 | 2.8 | 3.0 | 2.2 | 2.2 | 2.4 | 2.9 | 1.6 | 2.6 | 2.8 | 3.1 | 2.4 | 0.2 | 0.1 | 0.2 | 2.2 | 0.3 | 0.0 | 0.1 | 1.6 | 0.1 | 0.1 | 0.1 | 1.9 |
| | image | Inception-v3 | xxi | 99.8 | 91.1 | 3.4 | 9.4 | 99.4 | 84.7 | 3.0 | 2.9 | 46.0 | 57.1 | 2.1 | 2.5 | 93.0 | 64.0 | 0.0 | 0.6 | 98.8 | 73.3 | 0.0 | 0.0 | 13.2 | 20.4 | 0.0 | 0.0 |
| | | ResNet-101 | xxii | 99.3 | 69.8 | 2.9 | 11.2 | 98.5 | 53.1 | 2.6 | 4.0 | 34.1 | 9.4 | 3.1 | 2.2 | 90.5 | 42.0 | 0.1 | 2.0 | 96.2 | 28.5 | 0.0 | 0.1 | 9.8 | 3.5 | 0.0 | 0.0 |
| | | DenseNet-161 | xxiii | 99.1 | 65.6 | 2.1 | 5.1 | 97.7 | 36.0 | 2.5 | 3.0 | 43.9 | 36.1 | 2.3 | 3.0 | 89.5 | 30.5 | 0.1 | 0.0 | 96.4 | 11.7 | 0.0 | 0.0 | 15.0 | 13.6 | 0.0 | 0.0 |
| | | AlexNet | xxiv | 95.5 | 3.6 | 2.4 | 3.5 | 85.7 | 3.6 | 2.5 | 3.5 | 34.6 | 2.5 | 2.5 | 2.5 | 77.9 | 0.0 | 0.0 | 2.6 | 72.2 | 0.0 | 0.0 | 2.6 | 12.9 | 0.0 | 0.0 | 2.5 |

Table 17: The image encoder can generalize in some situations (some of columns ii, iv, vi, viii, x, xii, xiv, xvi, xviii, xx, xxii, and xxiv, for rows v–viii, xiii–xvi, and xxi–xxiv, are above chance). This, together with Tables 18 and 19, supports claim D.

| | | | original before (i) | original after joint pretraining (ii) | original after joint no pretraining (iii) | original after separate (iv) | 40 classes top-40 before (v) | 40 classes top-40 after joint pretraining (vi) | 40 classes top-40 after joint no pretraining (vii) | 40 classes top-40 after separate (viii) | bottom 960 before (ix) | bottom 960 after joint pretraining (x) | bottom 960 after joint no pretraining (xi) | bottom 960 after separate (xii) | original after before (xiii) | original after joint pretraining (xiv) | original after joint no pretraining (xv) | original after separate (xvi) | 1000 classes top-40 before (xvii) | 1000 classes top-40 after joint pretraining (xviii) | 1000 classes top-40 after joint no pretraining (xix) | 1000 classes top-40 after separate (xx) | bottom 960 after before (xxi) | bottom 960 after joint pretraining (xxii) | bottom 960 after joint no pretraining (xxiii) | bottom 960 after separate (xxiv) |
|---|---|---|---|---|---|---|---|---|---|---|---|---|---|---|---|---|---|---|---|---|---|---|---|---|---|---|
| their | EEG | Inception v3 | 2.6 | 3.0 | 2.5 | 55.5 | 2.2 | 2.6 | 2.5 | 55.5 | 2.5 | 3.1 | 2.5 | 2.3 | 0.1 | 0.1 | 0.1 | 55.5 | 0.1 | 0.1 | 0.1 | 55.5 | 0.1 | 0.1 | 0.1 | 2.3 |
| | | ResNet-101 | 2.3 | 3.0 | 2.6 | 54.7 | 2.8 | 2.4 | 2.4 | 54.7 | 2.4 | 3.1 | 2.5 | 2.3 | 0.0 | 0.1 | 0.1 | 54.7 | 0.0 | 0.0 | 0.1 | 54.7 | 0.1 | 0.1 | 0.1 | 2.3 |
| | | DenseNet-161 | 2.5 | 3.1 | 2.4 | 57.4 | 2.4 | 2.6 | 2.3 | 57.4 | 2.5 | 3.2 | 2.4 | 2.5 | 0.1 | 0.1 | 0.1 | 57.4 | 0.0 | 0.1 | 0.0 | 57.4 | 0.1 | 0.1 | 0.1 | 2.5 |
| | | AlexNet | 2.3 | 2.5 | 2.4 | 57.3 | 2.4 | 2.6 | 2.6 | 57.3 | 2.4 | 2.5 | 2.6 | 2.5 | 0.1 | 0.1 | 0.1 | 57.3 | 0.0 | 0.1 | 0.1 | 57.3 | 0.1 | 0.1 | 0.1 | 2.5 |
| | image | Inception v3 | 99.7 | 43.2 | 2.7 | 3.4 | 99.4 | 13.5 | 2.9 | 2.2 | 45.5 | 29.4 | 2.7 | 2.5 | 92.8 | 16.5 | 0.1 | 0.0 | 98.9 | 4.0 | 0.0 | 0.0 | 12.8 | 9.3 | 0.0 | 0.0 |
| | | ResNet-101 | 99.4 | 24.2 | 2.4 | 82.5 | 98.3 | 7.3 | 2.9 | 70.5 | 33.0 | 9.5 | 2.1 | 39.4 | 90.5 | 2.6 | 0.0 | 82.5 | 95.6 | 0.3 | 0.0 | 70.5 | 9.5 | 0.7 | 0.0 | 39.2 |
| | | DenseNet-161 | 99.2 | 7.2 | 2.4 | 23.3 | 97.9 | 4.4 | 2.3 | 3.5 | 42.7 | 3.8 | 2.6 | 2.6 | 89.4 | 0.1 | 0.0 | 12.1 | 95.8 | 0.0 | 0.0 | 0.1 | 14.2 | 0.0 | 0.1 | 0.0 |
| | | AlexNet | 95.8 | 2.6 | 2.7 | 18.7 | 85.8 | 2.6 | 2.7 | 19.1 | 34.1 | 2.6 | 2.4 | 2.4 | 78.4 | 0.0 | 0.3 | 18.7 | 72.5 | 0.0 | 0.3 | 19.1 | 12.5 | 0.0 | 0.0 | 2.4 |
| block | EEG | Inception v3 | 2.4 | 6.8 | 1.9 | 77.0 | 1.8 | 7.1 | 2.0 | 69.8 | 2.6 | 3.6 | 2.5 | 32.1 | 0.1 | 0.4 | 0.0 | 77.0 | 0.0 | 0.2 | 0.1 | 69.8 | 0.0 | 0.2 | 0.1 | 28.0 |
| | | ResNet-101 | 1.6 | 3.5 | 2.2 | 76.6 | 1.1 | 4.2 | 2.5 | 70.5 | 2.4 | 2.8 | 2.2 | 28.1 | 0.0 | 0.1 | 0.1 | 76.6 | 0.0 | 0.1 | 0.1 | 70.5 | 0.0 | 0.0 | 0.1 | 24.0 |
| | | DenseNet-161 | 1.4 | 4.1 | 2.0 | 77.0 | 1.2 | 3.8 | 1.9 | 72.2 | 2.5 | 3.5 | 2.7 | 25.9 | 0.1 | 0.1 | 0.0 | 77.0 | 0.0 | 0.0 | 0.0 | 72.2 | 0.1 | 0.1 | 0.1 | 22.3 |
| | | AlexNet | 2.9 | 4.1 | 2.9 | 76.5 | 2.5 | 4.2 | 2.2 | 70.0 | 2.0 | 3.4 | 2.5 | 27.8 | 0.0 | 0.1 | 0.1 | 76.4 | 0.0 | 0.3 | 0.2 | 70.0 | 0.0 | 0.0 | 0.0 | 24.2 |
| | image | Inception v3 | 99.8 | 96.4 | 3.0 | 9.3 | 99.4 | 94.7 | 3.1 | 2.9 | 46.0 | 46.8 | 2.4 | 2.5 | 93.0 | 75.1 | 0.1 | 0.5 | 98.8 | 87.0 | 0.0 | 0.0 | 13.2 | 11.7 | 0.0 | 0.0 |
| | | ResNet-101 | 99.3 | 75.0 | 2.4 | 11.0 | 98.5 | 68.5 | 2.7 | 3.5 | 34.1 | 14.7 | 2.5 | 3.7 | 90.5 | 56.4 | 0.0 | 2.0 | 96.2 | 52.4 | 0.0 | 0.1 | 9.8 | 0.2 | 0.0 | 0.0 |
| | | DenseNet-161 | 99.1 | 95.8 | 2.5 | 5.7 | 97.7 | 86.8 | 2.9 | 3.8 | 43.9 | 50.9 | 2.1 | 2.6 | 89.5 | 71.1 | 0.1 | 0.2 | 96.4 | 67.2 | 0.4 | 0.2 | 15.0 | 17.9 | 0.0 | 0.0 |
| | | AlexNet | 95.5 | 60.9 | 2.5 | 3.5 | 85.7 | 50.9 | 2.4 | 3.2 | 34.6 | 15.6 | 2.5 | 2.3 | 77.9 | 25.4 | 0.0 | 2.6 | 72.2 | 19.7 | 0.0 | 2.5 | 12.9 | 4.2 | 0.0 | 2.3 |
| randomized | EEG | Inception v3 | 2.6 | 2.5 | 2.5 | 1.6 | 2.6 | 2.9 | 2.6 | 2.5 | 3.0 | 2.2 | 2.7 | 1.6 | 0.1 | 0.1 | 0.1 | 1.5 | 0.1 | 0.1 | 0.0 | 2.5 | 0.2 | 0.1 | 0.2 | 1.4 |
| | | ResNet-101 | 2.6 | 2.4 | 2.8 | 2.1 | 2.8 | 2.7 | 2.9 | 2.3 | 2.6 | 2.7 | 3.0 | 1.8 | 0.0 | 0.1 | 0.1 | 1.9 | 0.0 | 0.1 | 0.2 | 2.3 | 0.1 | 0.2 | 0.1 | 1.6 |
| | | DenseNet-161 | 2.1 | 2.2 | 2.3 | 1.5 | 2.5 | 2.3 | 2.5 | 2.0 | 3.1 | 2.0 | 1.9 | 1.9 | 0.1 | 0.1 | 0.1 | 1.5 | 0.0 | 0.0 | 0.1 | 2.0 | 0.1 | 0.1 | 0.1 | 1.7 |
| | | AlexNet | 2.5 | 2.8 | 3.0 | 2.2 | 2.2 | 2.4 | 2.9 | 1.6 | 2.6 | 2.8 | 3.1 | 2.4 | 0.2 | 0.1 | 0.2 | 2.2 | 0.3 | 0.0 | 0.1 | 1.6 | 0.1 | 0.1 | 0.1 | 1.9 |
| | image | Inception v3 | 99.8 | 91.1 | 3.4 | 9.4 | 99.4 | 84.7 | 3.0 | 2.9 | 46.0 | 57.1 | 2.1 | 2.5 | 93.0 | 64.0 | 0.1 | 0.6 | 98.8 | 73.3 | 0.0 | 0.0 | 13.2 | 20.4 | 0.0 | 0.0 |
| | | ResNet-101 | 99.3 | 69.8 | 2.9 | 11.2 | 98.5 | 53.1 | 2.6 | 4.0 | 34.1 | 9.4 | 3.1 | 2.2 | 90.5 | 42.0 | 0.0 | 2.0 | 96.2 | 28.5 | 0.0 | 0.1 | 9.8 | 3.5 | 0.0 | 0.0 |
| | | DenseNet-161 | 99.1 | 65.6 | 2.1 | 5.1 | 97.7 | 36.0 | 2.5 | 3.0 | 43.9 | 36.1 | 2.3 | 3.0 | 89.5 | 30.5 | 0.1 | 0.0 | 96.4 | 11.7 | 0.0 | 0.0 | 15.0 | 13.6 | 0.0 | 0.0 |
| | | AlexNet | 95.5 | 3.6 | 2.4 | 3.5 | 85.7 | 3.6 | 2.5 | 3.5 | 34.6 | 2.5 | 2.5 | 2.5 | 77.9 | 0.0 | 0.0 | 2.6 | 72.2 | 0.0 | 0.0 | 2.6 | 12.9 | 0.0 | 0.0 | 2.5 |

Table 18: The EEG encoder can generalize in some situations on confounded data with separate training (some of columns iv, viii, xii, xvi, xx, and xxiv, for rows i–iv and ix–xii, are above chance). This, together with Tables 17 and 19, supports claim D.

| | | | original before | original after joint pretraining | original after joint no pretraining | original after separate | 40 classes top 40 before | top 40 after joint pretraining | top 40 after joint no pretraining | top 40 after separate | bottom 960 before | bottom 960 after joint pretraining | bottom 960 after joint no pretraining | bottom 960 after separate | original before | original after joint pretraining | original after joint no pretraining | original after separate | 1000 top 40 before | top 40 after joint pretraining | top 40 after joint no pretraining | top 40 after separate | bottom 960 before | bottom 960 after joint pretraining | bottom 960 after joint no pretraining | bottom 960 after separate |
|---|---|---|---|---|---|---|---|---|---|---|---|---|---|---|---|---|---|---|---|---|---|---|---|---|---|---|
| | | | i | ii | iii | iv | v | vi | vii | viii | ix | x | xi | xii | xiii | xiv | xv | xvi | xvii | xviii | xix | xx | xxi | xxii | xxiii | xxiv |
| their | EEG | Inception v3 | 2.6 | 3.0 | 2.5 | 55.5 | 2.2 | 2.6 | 2.5 | 55.5 | 2.5 | 3.1 | 2.5 | 2.3 | 0.1 | 0.1 | 0.1 | 55.5 | 0.1 | 0.1 | 0.1 | 55.5 | 0.1 | 0.1 | 0.1 | 2.3 |
| | | ResNet-101 | 2.3 | 3.0 | 2.6 | 54.7 | 2.8 | 2.4 | 2.4 | 54.7 | 2.4 | 3.1 | 2.5 | 2.3 | 0.0 | 0.1 | 0.1 | 54.7 | 0.0 | 0.0 | 0.1 | 54.7 | 0.1 | 0.1 | 0.1 | 2.3 |
| | | DenseNet-161 | 2.5 | 3.1 | 2.4 | 57.4 | 2.4 | 2.6 | 2.3 | 57.4 | 2.5 | 3.2 | 2.4 | 2.5 | 0.1 | 0.1 | 0.1 | 57.4 | 0.0 | 0.1 | 0.0 | 57.4 | 0.1 | 0.1 | 0.1 | 2.5 |
| | | AlexNet | 2.3 | 2.5 | 2.4 | 57.3 | 2.4 | 2.6 | 2.6 | 57.3 | 2.4 | 2.5 | 2.6 | 2.5 | 0.1 | 0.1 | 0.1 | 57.3 | 0.0 | 0.1 | 0.1 | 57.3 | 0.1 | 0.1 | 0.1 | 2.5 |
| | image | Inception v3 | 99.7 | 43.2 | 2.7 | 3.4 | 99.4 | 13.5 | 2.9 | 2.2 | 45.5 | 29.4 | 2.7 | 2.5 | 92.8 | 16.5 | 0.1 | 0.0 | 98.9 | 4.0 | 0.0 | 0.0 | 12.8 | 9.3 | 0.0 | 0.0 |
| | | ResNet-101 | 99.4 | 24.2 | 2.4 | 82.5 | 98.3 | 7.3 | 2.9 | 70.5 | 33.0 | 9.5 | 2.1 | 39.4 | 90.5 | 2.6 | 0.0 | 82.5 | 95.6 | 0.3 | 0.0 | 70.5 | 9.5 | 0.7 | 0.0 | 39.2 |
| | | DenseNet-161 | 99.2 | 7.2 | 2.4 | 23.3 | 97.9 | 4.4 | 2.3 | 3.5 | 42.7 | 3.8 | 2.6 | 2.6 | 89.4 | 0.1 | 0.0 | 12.1 | 95.8 | 0.0 | 0.0 | 0.1 | 14.2 | 0.0 | 0.1 | 0.0 |
| | | AlexNet | 95.8 | 2.6 | 2.7 | 18.7 | 85.8 | 2.6 | 2.7 | 19.1 | 34.1 | 2.6 | 2.4 | 2.4 | 78.4 | 0.0 | 0.3 | 18.7 | 72.5 | 0.0 | 0.3 | 19.1 | 12.5 | 0.0 | 0.0 | 2.4 |
| block | EEG | Inception v3 | 2.4 | 6.8 | 1.9 | 77.0 | 1.8 | 7.1 | 2.0 | 69.8 | 2.6 | 3.6 | 2.5 | 32.1 | 0.1 | 0.4 | 0.0 | 77.0 | 0.0 | 0.2 | 0.1 | 69.8 | 0.0 | 0.2 | 0.1 | 28.0 |
| | | ResNet-101 | 1.6 | 3.5 | 2.2 | 76.6 | 1.1 | 4.2 | 2.5 | 70.5 | 2.4 | 2.8 | 2.2 | 28.1 | 0.0 | 0.1 | 0.1 | 76.6 | 0.0 | 0.1 | 0.1 | 70.5 | 0.0 | 0.0 | 0.1 | 24.0 |
| | | DenseNet-161 | 1.4 | 4.1 | 2.0 | 77.0 | 1.2 | 3.8 | 1.9 | 72.2 | 2.5 | 3.5 | 2.7 | 25.9 | 0.1 | 0.1 | 0.0 | 77.0 | 0.0 | 0.0 | 0.0 | 72.2 | 0.1 | 0.1 | 0.0 | 22.3 |
| | | AlexNet | 2.9 | 4.1 | 2.9 | 76.5 | 2.5 | 4.2 | 2.2 | 70.0 | 2.0 | 3.4 | 2.5 | 27.8 | 0.0 | 0.1 | 0.1 | 76.4 | 0.0 | 0.3 | 0.2 | 70.0 | 0.0 | 0.0 | 0.0 | 24.2 |
| | image | Inception v3 | 99.8 | 96.4 | 3.0 | 9.3 | 99.4 | 94.7 | 3.1 | 2.9 | 46.0 | 46.8 | 2.4 | 2.7 | 93.0 | 75.1 | 0.1 | 0.5 | 98.8 | 87.0 | 0.0 | 0.0 | 13.2 | 11.7 | 0.0 | 0.0 |
| | | ResNet-101 | 99.3 | 75.0 | 2.4 | 11.0 | 98.5 | 68.5 | 2.7 | 3.5 | 34.1 | 14.7 | 2.5 | 3.7 | 90.5 | 56.4 | 0.1 | 2.0 | 96.2 | 52.4 | 0.0 | 0.1 | 9.8 | 0.2 | 0.0 | 0.1 |
| | | DenseNet-161 | 99.1 | 95.8 | 2.5 | 5.7 | 97.7 | 86.8 | 2.9 | 3.8 | 43.9 | 50.9 | 2.1 | 2.6 | 89.5 | 71.1 | 0.1 | 0.2 | 96.4 | 67.2 | 0.4 | 0.2 | 15.0 | 17.9 | 0.0 | 0.0 |
| | | AlexNet | 95.5 | 60.9 | 2.5 | 3.5 | 85.7 | 50.9 | 2.4 | 3.2 | 34.6 | 15.6 | 2.5 | 2.3 | 77.9 | 25.4 | 0.0 | 2.6 | 72.2 | 19.7 | 0.0 | 2.5 | 12.9 | 4.2 | 0.0 | 2.3 |
| randomized | EEG | Inception v3 | 2.6 | 2.5 | 2.5 | 1.6 | 2.6 | 2.9 | 2.6 | 2.5 | 3.0 | 2.2 | 2.7 | 1.6 | 0.1 | 0.1 | 0.1 | 1.5 | 0.1 | 0.1 | 0.0 | 2.5 | 0.2 | 0.1 | 0.2 | 1.4 |
| | | ResNet-101 | 2.6 | 2.4 | 2.8 | 2.1 | 2.8 | 2.7 | 2.9 | 2.3 | 2.6 | 2.7 | 3.0 | 1.8 | 0.0 | 0.1 | 0.1 | 1.9 | 0.0 | 0.1 | 0.2 | 2.3 | 0.1 | 0.2 | 0.1 | 1.6 |
| | | DenseNet-161 | 2.1 | 2.2 | 2.3 | 1.5 | 2.5 | 2.3 | 2.5 | 2.0 | 3.1 | 2.0 | 1.9 | 1.9 | 0.1 | 0.1 | 0.1 | 1.5 | 0.0 | 0.0 | 0.1 | 2.0 | 0.1 | 0.1 | 0.1 | 1.7 |
| | | AlexNet | 2.5 | 2.8 | 3.0 | 2.2 | 2.2 | 2.4 | 2.9 | 1.6 | 2.6 | 2.8 | 3.1 | 2.4 | 0.2 | 0.1 | 0.2 | 2.2 | 0.3 | 0.0 | 0.1 | 1.6 | 0.1 | 0.1 | 0.1 | 1.9 |
| | image | Inception v3 | 99.8 | 91.1 | 3.4 | 9.4 | 99.4 | 84.7 | 3.0 | 2.9 | 46.0 | 57.1 | 2.1 | 2.5 | 93.0 | 64.0 | 0.1 | 0.6 | 98.8 | 73.3 | 0.0 | 0.0 | 13.2 | 20.4 | 0.0 | 0.0 |
| | | ResNet-101 | 99.3 | 69.8 | 2.9 | 11.2 | 98.5 | 53.1 | 2.6 | 4.0 | 34.1 | 9.4 | 3.1 | 2.2 | 90.5 | 42.0 | 0.0 | 2.0 | 96.2 | 28.5 | 0.0 | 0.1 | 9.8 | 3.5 | 0.0 | 0.0 |
| | | DenseNet-161 | 99.1 | 65.6 | 2.1 | 5.1 | 97.7 | 36.0 | 2.5 | 3.0 | 43.9 | 36.1 | 2.3 | 3.0 | 89.5 | 30.5 | 0.1 | 0.0 | 96.4 | 11.7 | 0.0 | 0.0 | 15.0 | 13.6 | 0.0 | 0.0 |
| | | AlexNet | 95.5 | 3.6 | 2.4 | 3.5 | 85.7 | 3.6 | 2.5 | 3.5 | 34.6 | 2.5 | 2.5 | 2.5 | 77.9 | 0.0 | 0.0 | 2.6 | 72.2 | 0.0 | 0.0 | 2.6 | 12.9 | 0.0 | 0.0 | 2.5 |

Table 19: The EEG encoder cannot generalize on nonconfounded data with joint training with pretraining (all of columns ii, vi, x, xiv, xviii, and xxi, for rows xvii-xx, are at chance). This, together with Tables 17 and 18, supports claim D.

| | | | | 40 classes | | | | | | | | | | | |
|---|---|---|---|---|---|---|---|---|---|---|---|---|---|---|---|
| | | | | original | | | | top 40 | | | | bottom 960 | | | |
| | | | | before | after joint | | after separate | before | after joint | | after separate | before | after joint | | after separate |
| | | | | | pretraining | no pretraining | | | pretraining | no pretraining | | | pretraining | no pretraining | |
| | | | | i | ii | iii | iv | v | vi | vii | viii | ix | x | xi | xii |
| their | EEG | Inception v3 | i | 2.6 | 3.0 | 2.5 | 55.5 | 2.2 | 2.6 | 2.5 | 55.5 | 2.5 | 3.1 | 2.5 | 2.3 |
| | | ResNet-101 | ii | 2.3 | 3.0 | 2.6 | 54.7 | 2.8 | 2.4 | 2.4 | 54.7 | 2.4 | 3.1 | 2.5 | 2.3 |
| | | DenseNet-161 | iii | 2.5 | 3.1 | 2.4 | 57.4 | 2.4 | 2.6 | 2.3 | 57.4 | 2.5 | 3.2 | 2.4 | 2.5 |
| | | AlexNet | iv | 2.3 | 2.5 | 2.4 | 57.3 | 2.4 | 2.6 | 2.6 | 57.3 | 2.4 | 2.5 | 2.6 | 2.5 |
| | image | Inception v3 | v | 99.7 | 43.2 | 2.7 | 3.4 | 99.4 | 13.5 | 2.9 | 2.2 | 45.5 | 29.4 | 2.7 | 2.5 |
| | | ResNet-101 | vi | 99.4 | 24.2 | 2.4 | 82.5 | 98.3 | 7.3 | 2.9 | 70.5 | 33.0 | 9.5 | 2.1 | 39.4 |
| | | DenseNet-161 | vii | 99.2 | 7.2 | 2.4 | 23.3 | 97.9 | 4.4 | 2.3 | 3.5 | 42.7 | 3.8 | 2.6 | 2.6 |
| | | AlexNet | viii | 95.8 | 2.6 | 2.7 | 18.7 | 85.8 | 2.6 | 2.7 | 19.1 | 34.1 | 2.6 | 2.4 | 2.4 |
| block | EEG | Inception v3 | ix | 2.4 | 6.8 | 1.9 | 77.0 | 1.8 | 7.1 | 2.0 | 69.8 | 2.6 | 3.6 | 2.5 | 32.1 |
| | | ResNet-101 | x | 1.6 | 3.5 | 2.2 | 76.6 | 1.1 | 4.2 | 2.5 | 70.5 | 2.4 | 2.8 | 2.2 | 28.1 |
| | | DenseNet-161 | xi | 1.4 | 4.1 | 2.0 | 77.0 | 1.2 | 3.8 | 1.9 | 72.2 | 2.5 | 3.5 | 2.7 | 25.9 |
| | | AlexNet | xii | 2.9 | 4.1 | 2.9 | 76.5 | 2.5 | 4.2 | 2.2 | 70.0 | 2.0 | 3.4 | 2.5 | 27.8 |
| | image | Inception v3 | xiii | 99.8 | 96.4 | 3.0 | 9.3 | 99.4 | 94.7 | 3.1 | 2.9 | 46.0 | 46.8 | 2.4 | 2.5 |
| | | ResNet-101 | xiv | 99.3 | 75.0 | 2.4 | 11.0 | 98.5 | 68.5 | 2.7 | 3.5 | 34.1 | 14.7 | 2.5 | 3.7 |
| | | DenseNet-161 | xv | 99.1 | 95.8 | 2.5 | 5.7 | 97.7 | 86.8 | 2.9 | 3.8 | 43.9 | 50.9 | 2.1 | 2.6 |
| | | AlexNet | xvi | 95.5 | 60.9 | 2.5 | 3.5 | 85.7 | 50.9 | 2.4 | 3.2 | 34.6 | 15.6 | 2.5 | 2.3 |
| randomized | EEG | Inception v3 | xvii | 2.6 | 2.5 | 2.5 | 1.6 | 2.6 | 2.9 | 2.6 | 2.5 | 3.0 | 2.2 | 2.7 | 1.6 |
| | | ResNet-101 | xviii | 2.6 | 2.4 | 2.8 | 2.1 | 2.8 | 2.7 | 2.9 | 2.3 | 2.6 | 2.7 | 3.0 | 1.8 |
| | | DenseNet-161 | xix | 2.1 | 2.2 | 2.3 | 1.5 | 2.5 | 2.3 | 2.5 | 2.0 | 3.1 | 2.0 | 1.9 | 1.9 |
| | | AlexNet | xx | 2.5 | 2.8 | 3.0 | 2.2 | 2.2 | 2.4 | 2.9 | 1.6 | 2.6 | 2.8 | 3.1 | 2.4 |
| | image | Inception v3 | xxi | 99.8 | 91.1 | 3.4 | 9.4 | 99.4 | 84.7 | 3.0 | 2.9 | 46.0 | 57.1 | 2.1 | 2.5 |
| | | ResNet-101 | xxii | 99.3 | 69.8 | 2.9 | 11.2 | 98.5 | 53.1 | 2.6 | 4.0 | 34.1 | 9.4 | 3.1 | 2.2 |
| | | DenseNet-161 | xxiii | 99.1 | 65.6 | 2.1 | 5.1 | 97.7 | 36.0 | 2.5 | 3.0 | 43.9 | 36.1 | 2.3 | 3.0 |
| | | AlexNet | xxiv | 95.5 | 3.6 | 2.4 | 3.5 | 85.7 | 3.6 | 2.5 | 3.5 | 34.6 | 2.5 | 2.5 | 2.5 |

| | | | | 1000 classes | | | | | | | | | | | |
|---|---|---|---|---|---|---|---|---|---|---|---|---|---|---|---|
| | | | | original | | | | top 40 | | | | bottom 960 | | | |
| | | | | before | after joint | | after separate | before | after joint | | after separate | before | after joint | | after separate |
| | | | | | pretraining | no pretraining | | | pretraining | no pretraining | | | pretraining | no pretraining | |
| | | | | xiii | xiv | xv | xvi | xvii | xviii | xix | xx | xxi | xxii | xxiii | xxiv |
| their | EEG | Inception v3 | i | 0.1 | 0.1 | 0.1 | 55.5 | 0.1 | 0.1 | 0.1 | 55.5 | 0.1 | 0.1 | 0.1 | 2.3 |
| | | ResNet-101 | ii | 0.0 | 0.1 | 0.1 | 54.7 | 0.0 | 0.0 | 0.1 | 54.7 | 0.1 | 0.1 | 0.1 | 2.3 |
| | | DenseNet-161 | iii | 0.1 | 0.1 | 0.1 | 57.4 | 0.0 | 0.1 | 0.0 | 57.4 | 0.1 | 0.1 | 0.1 | 2.5 |
| | | AlexNet | iv | 0.1 | 0.1 | 0.1 | 57.3 | 0.0 | 0.1 | 0.1 | 57.3 | 0.1 | 0.1 | 0.1 | 2.5 |
| | image | Inception v3 | v | 92.8 | 16.5 | 0.1 | 0.0 | 98.9 | 4.0 | 0.0 | 0.0 | 12.8 | 9.3 | 0.0 | 0.0 |
| | | ResNet-101 | vi | 90.5 | 2.6 | 0.0 | 82.5 | 95.6 | 0.3 | 0.0 | 70.5 | 9.5 | 0.7 | 0.0 | 39.2 |
| | | DenseNet-161 | vii | 89.4 | 0.1 | 0.0 | 12.1 | 95.8 | 0.0 | 0.0 | 0.1 | 14.2 | 0.0 | 0.1 | 0.0 |
| | | AlexNet | viii | 78.4 | 0.0 | 0.3 | 18.7 | 72.5 | 0.0 | 0.3 | 19.1 | 12.5 | 0.0 | 0.0 | 2.4 |
| block | EEG | Inception v3 | ix | 0.1 | 0.4 | 0.0 | 77.0 | 0.0 | 0.2 | 0.1 | 69.8 | 0.0 | 0.2 | 0.1 | 28.0 |
| | | ResNet-101 | x | 0.0 | 0.1 | 0.1 | 76.6 | 0.0 | 0.1 | 0.1 | 70.5 | 0.0 | 0.0 | 0.1 | 24.0 |
| | | DenseNet-161 | xi | 0.1 | 0.1 | 0.0 | 77.0 | 0.0 | 0.0 | 0.0 | 72.2 | 0.1 | 0.1 | 0.1 | 22.3 |
| | | AlexNet | xii | 0.0 | 0.1 | 0.1 | 76.4 | 0.0 | 0.3 | 0.2 | 70.0 | 0.0 | 0.0 | 0.1 | 24.2 |
| | image | Inception v3 | xiii | 93.0 | 75.1 | 0.1 | 0.5 | 98.8 | 87.0 | 0.0 | 0.0 | 13.2 | 11.7 | 0.0 | 0.0 |
| | | ResNet-101 | xiv | 90.5 | 56.4 | 0.0 | 2.0 | 96.2 | 52.4 | 0.0 | 0.1 | 9.8 | 0.2 | 0.0 | 0.0 |
| | | DenseNet-161 | xv | 89.5 | 71.1 | 0.1 | 0.2 | 96.4 | 67.2 | 0.4 | 0.2 | 15.0 | 17.9 | 0.0 | 0.0 |
| | | AlexNet | xvi | 77.9 | 25.4 | 0.0 | 2.6 | 72.2 | 19.7 | 0.0 | 2.5 | 12.9 | 4.2 | 0.0 | 2.3 |
| randomized | EEG | Inception v3 | xvii | 0.1 | 0.1 | 0.1 | 1.5 | 0.1 | 0.1 | 0.0 | 2.5 | 0.2 | 0.1 | 0.2 | 1.4 |
| | | ResNet-101 | xviii | 0.0 | 0.1 | 0.1 | 1.9 | 0.0 | 0.1 | 0.2 | 2.3 | 0.1 | 0.2 | 0.1 | 1.6 |
| | | DenseNet-161 | xix | 0.1 | 0.1 | 0.1 | 1.5 | 0.0 | 0.0 | 0.1 | 2.0 | 0.1 | 0.1 | 0.1 | 1.7 |
| | | AlexNet | xx | 0.2 | 0.1 | 0.2 | 2.2 | 0.3 | 0.0 | 0.1 | 1.6 | 0.1 | 0.1 | 0.1 | 1.9 |
| | image | Inception v3 | xxi | 93.0 | 64.0 | 0.1 | 0.6 | 98.8 | 73.3 | 0.0 | 0.0 | 13.2 | 20.4 | 0.0 | 0.0 |
| | | ResNet-101 | xxii | 90.5 | 42.0 | 0.0 | 2.0 | 96.2 | 28.5 | 0.0 | 0.1 | 9.8 | 3.5 | 0.0 | 0.0 |
| | | DenseNet-161 | xxiii | 89.5 | 30.5 | 0.1 | 0.0 | 96.4 | 11.7 | 0.0 | 0.0 | 15.0 | 13.6 | 0.0 | 0.0 |
| | | AlexNet | xxiv | 77.9 | 0.0 | 0.0 | 2.6 | 72.2 | 0.0 | 0.0 | 2.6 | 12.9 | 0.0 | 0.0 | 2.5 |

Table 20: Classification accuracy is at chance after joint training without pretraining (columns iii vii, xi, xv, xix, and xxiii). This holds both for EEG classification and image classification, both for 40 classes and 1000 classes, both for 'their' data, 'block' data, and 'randomized' data, and whether using all components, just the 40 principal components, or the bottom 960 components. This suggests that their method completely breaks down when not provided with class information through pretraining, and further supports claim A and calls claim I into question.

| | | | | 40 classes — original | | | | 40 classes — top 40 | | | | 40 classes — bottom 960 | | | | 1000 classes — original | | | | 1000 classes — top 40 | | | | 1000 classes — bottom 960 | | | |
| | | | | before | after joint (pretr.) | after joint (no pretr.) | after separate | before | after joint (pretr.) | after joint (no pretr.) | after separate | before | after joint (pretr.) | after joint (no pretr.) | after separate | before | after joint (pretr.) | after joint (no pretr.) | after separate | before | after joint (pretr.) | after joint (no pretr.) | after separate | before | after joint (pretr.) | after joint (no pretr.) | after separate |
| | | | | i | ii | iii | iv | v | vi | vii | viii | ix | x | xi | xii | xiii | xiv | xv | xvi | xvii | xviii | xix | xx | xxi | xxii | xxiii | xxiv |
|---|---|---|---|---|---|---|---|---|---|---|---|---|---|---|---|---|---|---|---|---|---|---|---|---|---|---|---|
| their | EEG | Inception v3 | i | 2.6 | 3.0 | 2.5 | 55.5 | 2.2 | 2.6 | 2.5 | 55.5 | 2.5 | 3.1 | 2.5 | 2.3 | 0.1 | 0.1 | 0.1 | 55.5 | 0.1 | 0.1 | 0.1 | 55.5 | 0.1 | 0.1 | 0.1 | 2.3 |
| | | ResNe-101 | ii | 2.3 | 3.0 | 2.6 | 54.7 | 2.8 | 2.4 | 2.4 | 54.7 | 2.4 | 3.1 | 2.4 | 2.3 | 0.0 | 0.1 | 0.1 | 54.7 | 0.0 | 0.0 | 0.0 | 54.7 | 0.1 | 0.1 | 0.1 | 2.3 |
| | | DenseNet-161 | iii | 2.5 | 3.1 | 2.4 | 57.4 | 2.4 | 2.6 | 2.3 | 57.4 | 2.5 | 3.2 | 2.4 | 2.5 | 0.1 | 0.1 | 0.1 | 57.4 | 0.0 | 0.1 | 0.0 | 57.4 | 0.1 | 0.1 | 0.1 | 2.5 |
| | | AlexNet | iv | 2.3 | 2.5 | 2.4 | 57.3 | 2.4 | 2.6 | 2.6 | 57.3 | 2.4 | 2.5 | 2.6 | 2.5 | 0.1 | 0.1 | 0.1 | 57.3 | 0.0 | 0.1 | 0.1 | 57.3 | 0.1 | 0.1 | 0.1 | 2.5 |
| | image | Inception v3 | v | 99.7 | 43.2 | 2.7 | 3.4 | 99.4 | 13.5 | 2.9 | 2.2 | 45.5 | 29.4 | 2.7 | 2.5 | 92.8 | 16.5 | 0.1 | 0.0 | 98.9 | 4.0 | 0.0 | 0.0 | 12.8 | 9.3 | 0.0 | 0.0 |
| | | ResNe-101 | vi | 99.4 | 24.2 | 2.4 | 82.5 | 98.3 | 7.3 | 2.9 | 70.5 | 33.0 | 9.5 | 2.1 | 39.4 | 90.5 | 2.6 | 0.0 | 82.5 | 95.6 | 0.3 | 0.0 | 70.5 | 9.5 | 0.7 | 0.0 | 39.2 |
| | | DenseNet-161 | vii | 99.2 | 7.2 | 2.4 | 23.3 | 97.9 | 4.4 | 2.3 | 3.5 | 42.7 | 3.8 | 2.6 | 2.6 | 89.4 | 0.1 | 0.0 | 12.1 | 95.8 | 0.0 | 0.0 | 0.1 | 14.2 | 0.0 | 0.0 | 0.0 |
| | | AlexNet | viii | 95.8 | 2.6 | 2.7 | 18.7 | 85.8 | 2.6 | 2.7 | 19.1 | 34.1 | 2.6 | 2.4 | 2.4 | 78.4 | 0.0 | 0.3 | 18.7 | 72.5 | 0.0 | 0.3 | 19.1 | 12.5 | 0.0 | 0.0 | 2.4 |
| block | EEG | Inception v3 | ix | 2.4 | 6.8 | 1.9 | 77.0 | 1.8 | 7.1 | 2.0 | 69.8 | 2.6 | 3.6 | 2.5 | 32.1 | 0.1 | 0.4 | 0.0 | 77.0 | 0.0 | 0.2 | 0.1 | 69.8 | 0.0 | 0.2 | 0.1 | 28.0 |
| | | ResNe-101 | x | 1.6 | 3.5 | 2.2 | 76.6 | 1.1 | 4.2 | 2.5 | 70.5 | 2.4 | 2.8 | 2.2 | 28.1 | 0.0 | 0.1 | 0.1 | 76.6 | 0.0 | 0.1 | 0.1 | 70.5 | 0.0 | 0.0 | 0.0 | 24.0 |
| | | DenseNet-161 | xi | 1.4 | 4.1 | 2.0 | 77.0 | 1.2 | 3.8 | 1.9 | 72.2 | 2.5 | 3.5 | 2.7 | 25.9 | 0.1 | 0.1 | 0.0 | 77.0 | 0.0 | 0.0 | 0.0 | 72.2 | 0.1 | 0.1 | 0.0 | 22.3 |
| | | AlexNet | xii | 2.9 | 4.1 | 2.9 | 76.5 | 2.5 | 4.2 | 2.2 | 70.0 | 2.0 | 3.4 | 2.5 | 27.8 | 0.0 | 0.1 | 0.1 | 76.4 | 0.0 | 0.3 | 0.2 | 70.0 | 0.0 | 0.0 | 0.1 | 24.2 |
| | image | Inception v3 | xiii | 99.8 | 96.4 | 3.0 | 9.3 | 99.4 | 94.7 | 3.1 | 2.9 | 46.8 | 46.8 | 2.4 | 2.5 | 93.0 | 75.1 | 0.1 | 0.5 | 98.8 | 87.0 | 0.0 | 0.0 | 13.2 | 1.7 | 0.0 | 0.0 |
| | | ResNe-101 | xiv | 99.3 | 75.0 | 2.4 | 11.0 | 98.5 | 68.5 | 2.7 | 3.5 | 34.1 | 14.7 | 2.7 | 3.7 | 90.5 | 56.4 | 0.0 | 2.0 | 96.2 | 52.4 | 0.0 | 0.1 | 9.8 | 0.2 | 0.0 | 0.0 |
| | | DenseNet-161 | xv | 99.1 | 95.8 | 2.5 | 5.7 | 97.7 | 86.8 | 2.9 | 3.8 | 43.9 | 50.9 | 2.9 | 2.6 | 89.5 | 71.1 | 0.1 | 0.2 | 96.4 | 67.2 | 0.4 | 0.2 | 15.0 | 17.9 | 0.0 | 0.0 |
| | | AlexNet | xvi | 95.5 | 60.9 | 2.5 | 3.5 | 85.7 | 50.9 | 2.4 | 3.2 | 34.6 | 15.6 | 2.5 | 2.3 | 77.9 | 25.4 | 0.0 | 2.5 | 72.2 | 19.7 | 0.0 | 2.5 | 12.9 | 4.2 | 0.0 | 2.3 |
| randomized | EEG | Inception v3 | xvii | 2.6 | 2.5 | 2.5 | 1.6 | 2.6 | 2.9 | 2.6 | 2.5 | 3.0 | 2.2 | 2.7 | 1.6 | 0.1 | 0.1 | 0.1 | 1.5 | 0.1 | 0.1 | 0.0 | 2.5 | 0.2 | 0.1 | 0.2 | 1.4 |
| | | ResNe-101 | xviii | 2.6 | 2.4 | 2.8 | 2.1 | 2.8 | 2.7 | 2.9 | 2.3 | 2.6 | 2.7 | 3.0 | 1.8 | 0.0 | 0.1 | 0.1 | 1.9 | 0.0 | 0.1 | 0.2 | 2.3 | 0.1 | 0.2 | 0.1 | 1.6 |
| | | DenseNet-161 | xix | 2.1 | 2.2 | 2.3 | 1.5 | 2.5 | 2.3 | 2.5 | 2.0 | 3.1 | 2.0 | 1.9 | 1.9 | 0.1 | 0.1 | 0.1 | 1.5 | 0.0 | 0.0 | 0.1 | 2.0 | 0.1 | 0.1 | 0.1 | 1.7 |
| | | AlexNet | xx | 2.5 | 2.8 | 3.0 | 2.2 | 2.2 | 2.4 | 2.9 | 1.6 | 2.6 | 2.8 | 3.1 | 2.4 | 0.2 | 0.1 | 0.2 | 2.2 | 0.3 | 0.0 | 0.1 | 1.6 | 0.1 | 0.1 | 0.1 | 1.9 |
| | image | Inception v3 | xxi | 99.8 | 91.1 | 3.4 | 9.4 | 99.4 | 84.7 | 3.0 | 2.9 | 46.0 | 57.1 | 2.1 | 2.5 | 93.0 | 64.0 | 0.1 | 0.6 | 98.8 | 73.3 | 0.0 | 0.0 | 13.2 | 20.4 | 0.0 | 0.0 |
| | | ResNe-101 | xxii | 99.3 | 69.8 | 2.9 | 11.2 | 98.5 | 53.1 | 2.6 | 4.0 | 34.1 | 9.4 | 3.1 | 2.2 | 90.5 | 42.0 | 0.0 | 2.0 | 96.2 | 28.5 | 0.0 | 0.1 | 9.8 | 3.5 | 0.0 | 0.0 |
| | | DenseNet-161 | xxiii | 99.1 | 65.6 | 2.1 | 5.1 | 97.7 | 36.0 | 2.5 | 3.0 | 43.9 | 36.1 | 2.3 | 3.0 | 89.5 | 30.5 | 0.1 | 0.0 | 96.4 | 11.7 | 0.0 | 0.0 | 15.0 | 13.6 | 0.0 | 0.0 |
| | | AlexNet | xxiv | 95.5 | 3.6 | 2.4 | 3.5 | 85.7 | 3.6 | 2.5 | 3.5 | 34.6 | 2.5 | 2.5 | 2.5 | 77.9 | 0.0 | 0.0 | 2.6 | 72.2 | 0.0 | 0.0 | 2.6 | 12.9 | 0.0 | 0.0 | 2.5 |

Table 21: With only two exceptions, separate training gets to a lower loss than joint training with pretraining (compare columns vii, viii, and ix to i, ii, and iii, respectively). This suggests that there is a point within the representational-capacity space of their model and loss function that achieves lower loss than achieved by their joint-training procedure. The joint-training procedure could have achieved it, it just didn't. Since that point was achieved with separate training, the resulting EEG encodings do not have any image information and the resulting image encodings do not have any brain-activity information. This supports claim B and calls claims II-IV into question.

| | | joint | | | | | | separate | | |
| | | pretraining | | | no pretraining | | | their | block | randomized |
| | | their | block | randomized | their | block | randomized | | | |
| | | i | ii | iii | iv | v | vi | vii | viii | ix |
|---|---|---|---|---|---|---|---|---|---|---|
| Inception v3 | i | 1.303 | 1.098 | 0.932 | 1.074 | 0.938 | 0.919 | 1.150 | 1.027 | 1.104 |
| ResNet-101 | ii | 1.155 | 0.925 | 1.019 | 0.963 | 1.006 | 1.080 | 0.885 | 0.936 | 0.884 |
| DenseNet-161 | iii | 1.027 | 1.080 | 1.047 | 0.988 | 0.919 | 0.929 | 1.007 | 0.982 | 1.029 |
| AlexNet | iv | 1.417 | 1.142 | 0.981 | 1.136 | 1.055 | 1.034 | 0.925 | 0.956 | 0.935 |

Table 22: With only three exceptions, joint training without pretraining gets to a lower loss than with pretraining (compare columns iv, v, and vi to i, ii, and iii, respectively). Yet classification accuracy is at chance after joint training without pretraining (columns iii vii, xi, xv, xix, and xxiii of Tables 2 and 3). This suggests that joint training without pretraining can memorize the training set yet fail to generalize at all, and further supports claim A and calls claim I into question.

| | | joint | | | | | | separate | | |
| | | pretraining | | | no pretraining | | | their | block | randomized |
| | | their | block | randomized | their | block | randomized | | | |
| | | i | ii | iii | iv | v | vi | vii | viii | ix |
|---|---|---|---|---|---|---|---|---|---|---|
| Inception v3 | i | 1.303 | 1.098 | 0.932 | 1.074 | 0.938 | 0.919 | 1.150 | 1.027 | 1.104 |
| ResNet-101 | ii | 1.155 | 0.925 | 1.019 | 0.963 | 1.006 | 1.080 | 0.885 | 0.936 | 0.884 |
| DenseNet-161 | iii | 1.027 | 1.080 | 1.047 | 0.988 | 0.919 | 0.929 | 1.007 | 0.982 | 1.029 |
| AlexNet | iv | 1.417 | 1.142 | 0.981 | 1.136 | 1.055 | 1.034 | 0.925 | 0.956 | 0.935 |

