# OpenReview forum: "Joint Training Does Not Transfer Information between EEG and Image Classifiers"
_ICLR.cc/2024/Conference — ICLR 2024 Conference Withdrawn Submission_

### Official Review · Reviewer_MkfK · 2023-10-31

**Soundness:** 2 fair
**Presentation:** 1 poor
**Contribution:** 1 poor
**Rating:** 1
**Confidence:** 5

**Summary:**

This article is a rebuttal to the paper previously published in PAMI titled "Decoding brain representations by multimodal learning of neural activity and visual features."( Palazzo et al., 2021) This paper argues that they have successfully refuted all the points made in the prior PAMI paper.

**Strengths:**

The perspectives and viewpoints of this paper are sharp and critical. It aims to challenge the PAMI paper (Palazzo et al., 2021) to prompt researchers in the machine learning community to reflect on everything from data collection to research paradigms when studying EEG-related problems.

**Weaknesses:**

### Clarity of Presentation

The paper's presentation is not optimal. While its aim is to refute all points made in (Palazzo et al., 2021), it does not systematically or clearly introduce the content from (Palazzo et al., 2021). A rigorous critique requires relevant equations and plots for clear justification or intuitive understanding, both of which are absent in the paper. Tables 2 and 3, in particular, are challenging to decipher.

### Audience Suitability and Depth of Analysis
The content might not be best suited for the ICLR audience. While the act of rebuttal is acceptable, the paper falls short in offering meaningful insights, explanations, or suggested improvements. It lacks adequate context, making it burdensome for a reviewer to consult another paper for making a fair judgment on this one. Claims such as "the dataset and data collection methods are fundamentally flawed" lack the requisite evidence to be convincing. Moreover, multiple works (e.g., Bai et al., 2023; Zeng et al., 2023; and others) are critiqued without adequate explanations or justifications.

**Questions:**

1. Can the authors clarify why they chose not to include a comprehensive overview or detailed context of the (Palazzo et al., 2021) paper, especially when the core of your paper relies on its refutation?
2.  How do you believe this paper aligns with the expectations of the ICLR community?
3. The author stated that "the dataset and data collection methods are fundamentally flawed." Can you provide specific evidence or examples to substantiate this claim? What concrete proof can be offered to validate this significant critique?

---

### Official Review · Reviewer_MS4c · 2023-10-31

**Soundness:** 1 poor
**Presentation:** 2 fair
**Contribution:** 1 poor
**Rating:** 1
**Confidence:** 5

**Summary:**

The authors present an analysis of a paper on EEG signal classification, raising doubts on the validity of results and providing experiments of their own to support their claims.

**Strengths:**

The paper does not have a significant methodological content, hence it is difficult to highlight its strengths. Of course, initiatives aimed at shedding light on alleged flaws in published works, when pursued in a constructive and respectable way, are welcome by the scientific community.

**Weaknesses:**

In order to confute Palazzo et al.’s work, the authors apply PCA to show that most of the variance of image and EEG embeddings is explained by the first 40 principal components. Based on my understanding of the work, I do not agree with the conclusions drawn by the authors. The fact that the first 40 PCs are able to describe image embeddings is expected, since the image encoder in Palazzo et al. is pre-trained on ImageNet, which includes the 40 classes used in the dataset: it is reasonable that features extracted by the encoder linearly separate the target classes. The fact that the first 40 PCs are able to describe EEG embeddings sounds to me as a confirmation that the EEG encoder effectively extracts neural activity patterns that correlate well with image content. The authors imply that this correlation is a consequence of the “confounded” nature of the EEG dataset: however, this statement seems to be subject of debate (see Li et al., 2021 and Ahmed et al., 2022, from the paper; and [A] below). It should also be noted that the authors claim that “Several independent lines of research have completely refuted a large body of completely flawed work”, although — based on the references — these “independent lines of research” exhibit a significant coauthorship overlap.

From these considerations, I do not believe that the paper makes a strong point in proving its case. Additionally, besides my concerns on the soundness of the work, in my opinion the paper is not suitable for ICLR, since it does not provide any methodological contribution, and I do not envisage a lot of interest but for a very specific niche of the audience. I think this kind of work is more suitable as a comment paper to the relevant prior publication.

[A] Palazzo, S., Spampinato, C., Schmidt, J., Kavasidis, I., Giordano, D., & Shah, M. (2020). Correct block-design experiments mitigate temporal correlation bias in EEG classification. arXiv preprint arXiv:2012.03849.

**Questions:**

I do not have any questions for the authors.

---

### Official Review · Reviewer_rJjg · 2023-11-01

**Soundness:** 3 good
**Presentation:** 2 fair
**Contribution:** 2 fair
**Rating:** 1
**Confidence:** 5

**Summary:**

The manuscript discusses the claims that are made in a recent paper (Palazzo et al. (2021)) on the task of EEG-based object recognition from image stimuli. In the publication of Palazzo et al. (2021), a joint-training method is proposed, and the publication's authors claim that brain activity information is transferred to a visual encoder and visual information is transferred to an EEG encoder. Moreover, the publication of Palazzo et al. (2021) claims that class label information is not used during this training process. The manuscript investigates whether the claims of Palazzo et al. (2021) hold true, performing experiments that invalidate them. The experiments are performed in three different datasets, using four different image encoder architectures and different training schemes.

The contributions are the following:
1) the authors show that pretrained image encoders contain close to perfect class information
2) EEG and image encoder models have learning capacity to memorize one-hot encodings and can be found with separate training
3) joint training degrades performance of visual encoders
4) EEG encoders perform at chance on non-confounded data

Overall, the manuscript refutes the claims of Palazzo et al. (2021), yet the authors do not claim that EEG-based object recognition is not possible.

References:
Palazzo et al., "Decoding brain representations by multimodal learning of neural activity and visual features", IEEE TPAMI 2021

**Strengths:**

A strength of this work is that the validity of the method proposed in (Palazzo et al. (2021)) has not been extensively studied in previous works. In fact Ahmed et al. (2021) have only investigated the effectiveness of the architecture proposed in (Palazzo et al. (2021)), without studying the effectiveness of the joint-training methodology. Moreover, the authors demonstrate that the method of Palazzo et al. (2021) does not exhibit its claims, whether applied to confounded (i.e. experimental data collection designs that may lead to high decoding accuracies through spurious artifacts) or nonconfounded data.

The analysis that the authors perform and present in Table 1, to explore whether visual and EEG encodings carry class label information, is interesting. The main findings of the manuscript are shown in Tables 2 and 3, making it clear that the joint training scheme of Palazzo et al. (2021) actually uses class label information originating from the visual encodings. Furthermore, it is shown that EEG encodings perform above chance-level accuracies only on confounded data. The fact that the experiments are performed with four different image encoder models, (Inception v3, ResNet-101, DenseNet-161, AlexNet), with and without pretraining on ImageNet, adds more credibility to the experiments.

Overall, indeed the manuscript helps towards obtaining a more complete understanding on the reasons why the claims made in Palazzo et al. (2021) are not valid, highlighting previously unexplored aspects. I find all of the results presented in this manuscript to be valid, however I would be more confident if I could additionaly read a clear description of the EEG preprocessing settings along with the reasoning behind them.

References:
Ahmed et al., "Object classification from randomized EEG trials", IEEE CVPR 2021
Li et al., "The perils and pitfalls of block design for EEG classification experiments", IEEE TPAMI 2021

**Weaknesses:**

The related work of the manuscript could be expanded, by discussing more recent works and datasets in the area of EEG-based object recognition (e.g. Gifford et al. (2022))
References:
Gifford AT, Dwivedi K, Roig G, Cichy RM. A large and rich EEG dataset for modeling human visual object recognition. NeuroImage. 2022 Dec 1;264:119754.

Dataset descriptions are not sufficient, hence the manuscript is not self-contained. Details which may be found by reading the relevant references, are missing (e.g. why are data only from subject 6 picked?). Such details should be described at some level within the manuscript. Moreover, I believe that the names which are used to denote the datasets throughout the manuscript ("theirs", "block", "randomized") could be changed so as to become more descriptive. E.g. the name "theirs" does not help the reader to understand where does it refer to. Also the name "block" can be confusing to some readers, as actually both (Palazzo et al. (2021)) and (Ahmed et al. (2021)) datasets have a block design. EEG signal preprocessing details are not mentioned, which also renders the manuscript less self-contained.

Section 2 (Method) could be organized in a more clear way. At its current form, it is difficult for a reader to follow the flow of this section. Some low level details (e.g. numbers related to the PCA analysis, numbers of classes in ImageNet and the EEG datasets, which cases count as misclassifications etc.) are presented in this Section, making it difficult to obtain a high level understanding of each experiment, e.g. the motivation behind it, an abstract experimental setup, the expected conclusions etc.

Section 3 (Results) needs to be presented in a more organized and coherent way. At its current form, this section does not enable a gradual understanding of the experimental analysis, as it simply describes the result of each Table, without allowing for a more gentle introduction to each experiment.

There are several presentation issues on the Tables of the manuscript. For example, regardings Tables 1, 2 and 3, instead of matching row IDs to particular image encoding models and particular datasets, it would be better to directly mention these information inside the Tables (instead of providing instructions on how to interpret these information through the caption). There are two errors in the caption of Table 2 (error 1: "Rows i-vii 'their'" should actually be "Rows i-viii 'their'", error 2: "Rows v-vii" should actually be "Rows v-viii").

Despite the interesting takeaways of this work, I find the value that it would bring to the community of ICLR to be below the standards. Several issues related to the method of Palazzo et al. (2021) have been discussed in previous works (e.g. Ahmed et al. (2021) and Li et al. (2021)). Raising additional issues about the method of Palazzo et al. (2021) contributes to an already opened discussion, yet it does not positively show any way forward. The authors state that "We make no claim that it is difficult or impossible to perform the central task attempted by Palazzo et al. (2021)", however the line of research that the manuscript follows, and further explores, would highly benefit from additional methodological contributions that could yield improvements on the studied task.

There are some typos, grammar, syntax and other issues throughout the manuscript, e.g.:
- " 'their,' " (caption of Table 1)
- "and!xxi-xxiv" (caption of Table 3)
- The authors should replace the word "falsified" (page 9), with a more proper one.

References:
Ahmed et al., "Object classification from randomized EEG trials", IEEE CVPR 2021
Li et al., "The perils and pitfalls of block design for EEG classification experiments", IEEE TPAMI 2021

**Questions:**

Among the results that are presented in Tables 2 and 3, there are particular columns where results on the 1000-class ImageNet classification problem are shown, using EEG encodings. However, all three datasets (namely "theirs", "block", "randomized"), contain EEG trials corresponding to image stimuli from 40 ImageNet classes. Hence it is unclear to what experimental setting what do these results correspond. Could the authors clarify this?

The authors train EEG and image encoders in two ways: 1) joint training and 2) separate training with MSE loss against one-hot class encodings. However, the authors do not explicitly state the motivation for choosing an MSE loss function for the separate training (e.g. did the authors consider using a standard cross-entropy loss to learn the one-hot encodings?).

Why did the authors choose to focus on the first 40 principal components (i.e. a fixed number of components) in the analysis that was performed in the encodings? Wouldn't it be better to perform a sweep over the number of principal components to show how does the explained variance change as more and more components are kept?

---

### Official Review · Reviewer_LQpL · 2023-11-01

**Soundness:** 2 fair
**Presentation:** 3 good
**Contribution:** 2 fair
**Rating:** 5
**Confidence:** 4

**Summary:**

This paper is about a fascinating topic, namely the joint information of stimulus (X=images), brain responses when viewing that image (Y=EEG), and relevant semantics for a human observer (C=classes/objects).
The present paper depards from a sequence of papers that started in 2017 (Palazzo et al. 2017, 2020, 2022). Note, the earlier papers have already been quite severely critized due to a problematic experimental design inducing strong correlations between test and training set (Ahmed et al. 2020, 2021).

The central contribution in the present paper is to add new critigue based on claims that there are non-class mediated features/dependencies between images and EEG (i.e. not mediated by labels)-
The question is never formalized in the present work but can be stated something like:

Question:   p(X,Y|C) = p(X|C)p(Y|C)?
In words: Are there dependencies between images and EEG that are not mediated by the class labels?

The present paper re-analyses the data using multiple learned representations of X and Y, and the conclusion is that the conditional independency seems to be confirmed.

I think this is a valid contribution and as such relevant for ICLR.

My main reservation is that the paper is not communicating the problem, the design of the investigation and the conclusion in a form that makes it possible to weigh the evidence.

For example:
1) Missing formalization.  The claims are described verbally (I-IV and A-D) but not formalized as mathematical/statistical relations that can assist understanding and guide experimental design.

2) Unclear design. Section 2 Method is a list of heuristics lacking justification - the "why" -what are steps aimed at in terms of evidence collection? I think the design is careful and makes up for the basic problems pointed out in the Palazzo-papers by Ahmed et al.

3) Weight of the evidence. Although the experiments are quite careful and detailed described it is impossible to judge the significance, we are missing estimates of uncertainty.
We learn: "It is highly unlikely that the suboptimal EEG model that they produce by joint training has any image
information, the suboptimal image model that they produce by joint training has any brain-activity
information, and neither suboptimal model produced by joint training has anything other than class
information, even when run on nonconfounded data."

How unlikely?

These observations lead me to express my reservations on the relevance for ICLR: This is a really interesting research question but needs more work to conclude.

**Strengths:**

The question raised is really interesting.
The experimental design seems relevant and is quite detailed described.
I do believe the conclusion could be correct for the given data and models.

**Weaknesses:**

The basic question needs a mathematical/statistical formulation that can guide experimental design and give context for the conclusion.
The experimental design needs better justification - the "why" for each step.
Missing uncertainty estimates. Please clarify to allow the reader to weigh the evidence.

**Questions:**

The basic question needs a mathematical/statistical formulation that can guide experimental design and give context for the conclusion.
The experimental design needs better justification - the "why" for each step.
Missing uncertainty estimates. Please clarify to allow the reader to weigh the evidence.

---

### Official Review · Reviewer_KoK5 · 2023-11-08

**Soundness:** 3 good
**Presentation:** 3 good
**Contribution:** 1 poor
**Rating:** 5
**Confidence:** 1

**Summary:**

The manuscript critically examines a previously proposed method by Palazzo et al., 2021, that jointly trains two neural networks: one for encoding images and another for encoding EEG data from subjects observing those images. Palazzo et al. claimed their method generates a joint embedding of both data types without relying on class labels, aiming to extract a deeper understanding of the relation between the two data modalities. They also assert that this method allows for bidirectional information transfer between the image and EEG encoders.

The current manuscript aims to disprove the primary claims (I-IV) made by Palazzo et al., arguing that the joint training method does not support these claims, regardless of whether the data is confounded. The authors introduce novel observations to further refute the assertions made in the original study.

**Strengths:**

1.  The paper is commendably structured, systematically dissecting each claim made by Palazzo et al., ensuring that every aspect of the original paper's arguments is evaluated.
2. The authors conducted extensive experiments to back their counterclaims. Their empirical analysis adds weight to their arguments.
3. The authors shared the raw data and code, to enhance the credibility of their work and ensure transparency, and encourage reproducibility in the research community. This openness enhances the trustworthiness of their work and facilitates peer validation.

**Weaknesses:**

1. The focused nature of the manuscript, primarily critiquing a specific journal paper, raises questions about its suitability for a broader conference audience.

2. The manuscript's contribution to advancing general EEG decoding techniques remains ambiguous.

3. The manuscript's novelty is somewhat challenging to ascertain, as it primarily focuses on validating the work of others. The authors might consider adding a general discussion section that elaborates on any novel insights or methodologies that their validation efforts have yielded, thereby contributing to the field's body of knowledge.

**Questions:**

N/A